# LungTTA: Text-to-Audio Generation of Synthetic Lung Sounds for Respiratory Health

## Abstract

Respiratory audio analysis is still limited by data scarcity, as real recordings are difficult to collect and often involve privacy and clinical constraints, which makes it harder to train robust machine learning models. We introduce LungTTA, a text-to-audio framework based on a latent diffusion model, which generates respiratory sounds such as cough, breathing, and phonation from structured prompts. The model is fine-tuned on 116,660 publicly available recordings and includes a retrieval-based memory component together with watermarking for traceability. We evaluate the generated audio using Fréchet Audio Distance (FAD), Kullback–Leibler (KL) divergence, and Inception Score (IS), and also introduce PRISM (Pulmonary Respiratory Integrity & Similarity Metric) ~~a domain aware metric~~ designed to capture respiratory signal structure. LungTTA achieves a FAD of 2.72, KL of 0.50, IS of 1.22, and PRISM of 0.23, compared to Stable Audio Open (6.73, 0.67) for FAD and KL, Make-An-Audio (1.54) for IS, and RespAgent (0.24) for PRISM. In human evaluation, LungTTA achieves 80.91 (Overall Quality, OVL) and 75.13 (Relevance to Text, REL), compared to RespAgent (59.27, 58.97) and EZAudio (55.24, 52.69), while expert assessment yields 58.33 (OVL), 44.44 (REL), and 38.89 (Clinical Relevance for Assessment, CRA), compared to RespAgent (56.94, 43.06, 36.11) and EZAudio (36.11, 29.17, 33.33). ~~In a downstream COVID-19 cough classification task, LungTTA improves performance under a VGGish-based setting, increasing AUC from 0.7331 (no augmentation) and 0.7631 (classical augmentation) to 0.7701 using LungTTA.~~ **In downstream evaluation, LungTTA achieved an AUC of 0.7701 for COVID-19 cough classification, compared with 0.7331 without augmentation and 0.7631 with classical augmentation. In a separate asthma-classification experiment, LungTTA achieved an AUC of 0.5852, compared with 0.5658 using classical augmentation.** ~~These results demonstrate that LungTTA-generated synthetic respiratory audio can be used as an effective data augmentation method.~~ **These findings suggest that LungTTA-generated respiratory audio can support controlled data-augmentation experiments.**

**Keywords:** text-to-audio generation, respiratory sound synthesis, latent diffusion models, data augmentation, pulmonary health, synthetic medical audio

## 1 Introduction

Pulmonary diseases such as Chronic Obstructive Pulmonary Disease (COPD) and asthma remain among the leading causes of morbidity and mortality worldwide, motivating the development of reliable tools for diagnosis and continuous monitoring (Soriano et al., 2017; Li et al., 2020). Audio-based approaches, including lung-sound analysis and machine learning classification, have shown that coughs, breathing sounds, and phonation signals can be used as acoustic biomarkers for respiratory health assessment (Mosuily et al., 2023; Nemati et al., 2022). However, collecting high-quality respiratory audio data is constrained by privacy and ethical requirements, dependence on clinical recording equipment, and the difficulty of recruiting representative patient cohorts. As a result, many publicly available datasets are limited in both size and diversity, which restricts the robustness and generalization of downstream machine learning models (Xia et al., 2022).

Synthetic audio generation has therefore been explored as a way to mitigate data scarcity. Recent text-to-audio (TTA) models such as AudioLDM2 and Stable Audio Open generate high-quality general-purpose audio conditioned on text descriptions (Liu et al., 2023; Majumder et al., 2024; Evans et al., 2025). However, these models are trained in generalized audio domains and do not explicitly model the temporal structure, frequency characteristics, or task-specific requirements of respiratory sounds. Conventional augmentation methods, such as time shifting, noise injection, and filtering, operate directly on existing recordings and therefore tend to produce variations of the same signals rather than genuinely new respiratory patterns (Xia et al., 2022). GAN-based approaches attempt to address this by generating new samples, but in practice they can be difficult to train and may produce unstable or inconsistent outputs in respiratory audio settings (Chakraborty et al., 2024). More recently, diffusion-based models have been adopted as a more stable alternative, with the ability to model complex audio distributions at higher fidelity (Feng et al., 2024). We introduce LungTTA, a text-to-audio generative framework for synthetic respiratory sound generation. LungTTA uses prompt-based conditioning to control attributes such as sound type, age, and smoking status, and is trained on a curated collection of publicly available respiratory datasets spanning cough, breathing, vowel phonation, and speech-based tasks. The goal is not to replace clinical data collection, but to provide a controlled and reproducible method for generating additional training data in data-scarce settings. All generated audio is embedded with a digital watermark that enables identification of synthetic samples.

We summarize the specific contributions of this work below.

- We present **LungTTA**, ~~a domain-specific text-to-audio framework for respiratory sound synthesis, with a unified conditioning formulation $Z_{\text{cond}} = [H; Z_{\text{mem}}; Z_{\text{meta}}]$ that integrates prompt semantics, retrieval-based exemplar priors, and structured metadata.~~ **a respiratory-domain adaptation of the Stable Audio Open text-to-audio backbone for synthetic respiratory sound generation. LungTTA inherits the pretrained audio autoencoder, T5-based text encoder, and diffusion transformer backbone, while introducing respiratory-specific prompt construction, metadata conditioning, retrieval-guided memory conditioning, and watermarking for traceability.**

- We formulate a metadata-grounded prompt construction and retrieval-guided conditioning pipeline that enables reproducible control of clinically relevant attributes.

- We introduce ~~**PRISM (Pulmonary Respiratory Integrity & Similarity Metric),** a domain-aware evaluation metric~~ **PRISM Pulmonary Respiratory Integrity & Similarity Metric** that complements standard metrics (FAD, KL, IS) by quantifying respiratory signal structure ~~using waveform-level features~~.

- We perform evaluation using objective metrics, human and expert listening studies, and ~~a downstream COVID-19 cough classification task~~ **a downstream COVID-19 cough and asthma classification tasks**, showing that LungTTA improves synthetic-data quality and downstream augmentation performance under controlled experimental settings.

## 2 Related Work

Respiratory audio modeling remains challenging due to limited datasets and variability in recording conditions, devices, and annotation quality. Although such variability can in principle support generalization, inconsistencies across datasets often make it difficult to develop robust and reliable models in practice (Xia et al., 2022; Niizumi et al., 2025). In addition, reliance on real patient recordings introduces privacy, ethical, and logistical constraints, which limit large-scale data collection and reinforce data scarcity as a key bottleneck in this domain. Early work addressed these limitations through classical augmentation techniques, applying signal-level transformations such as time shifting, noise injection, and filtering to expand training data. These methods are useful for regularization, but they primarily generate variations of existing recordings and provide limited control over clinically meaningful respiratory structure (Xia et al., 2022). GAN-based methods, such as CoughGAN ~~Ramesh et al. (2020)~~ **(Ramesh et al., 2020)**, introduced learned synthesis of respiratory sounds, but in practice they are often affected by unstable training, mode collapse,

and reduced diversity, which limits their suitability for high-fidelity medical audio generation (Chakraborty et al., 2024).

More recently, diffusion-based models have become a strong alternative for high-quality audio generation. AudioLDM ~~Liu et al. (2023)~~ **(Liu et al., 2023)** demonstrated effective text-to-audio synthesis using latent diffusion, while AudioLDM 2 ~~Liu et al. (2024)~~ **(Liu et al., 2024)** extended this framework to a unified setting covering speech, music, and general audio.

Additional systems, including EZAudio ~~Feng et al. (2024)~~ **(Feng et al., 2024)**, Make-An-Audio ~~Huang et al. (2023)~~ **(Huang et al., 2023)**, and Stable Audio Open ~~Evans et al. (2025)~~ **(Evans et al., 2025)**, further improve scalability and controllability in general audio generation, and speech-focused diffusion models such as VoiceLDM ~~Lee et al. (2024)~~ **(Lee et al., 2024)** similarly demonstrate strong performance in text-to-speech tasks.

OPERA ~~Zhang et al. (2024)~~ **(Zhang et al., 2024)** introduced a foundation-model framework for respiratory audio, emphasizing the importance of curated datasets and task-specific evaluation, while more recent work such as RespAgent ~~Zhang et al. (2026)~~ **(Zhang et al., 2026)** proposed a multimodal system that combines diagnosis and synthesis in a closed-loop framework. However, RespAgent does not operate as a standalone text-to-audio model. Its generation relies on both diagnostic context and reference audio representations derived from BEATs tokens, rather than text alone. In practice, this means that synthesis is guided by both prompts and reference audio. In contrast, LungTTA is designed to generate respiratory sounds directly from text prompts, covering cough, breathing, and phonation signals without requiring reference audio. This difference places LungTTA alongside general text-to-audio models, while remaining focused on respiratory data. A comparison of these approaches is shown in Table 1.

Table 1: Positioning of LungTTA relative to general audio generation models, respiratory-specific ~~systems~~ **representation and synthesis systems**, and augmentation methods.

| Method | Training Data | Backbone | Task | Resp.-Specific | Text-to-Audio |
|---|---|---|---|---|---|
| *General Audio Generation Models* | | | | | |
| AudioLDM (Liu et al., 2023) | General audio | Latent diffusion | Text-to-audio | ✗ | ✓ |
| VoiceLDM (Lee et al., 2024) | ~~General audio (speech)~~ General audio / speech | Latent diffusion | Text-to-speech | ✗ | ✓ |
| AudioLDM 2 (Liu et al., 2024) | General audio | Latent diffusion | Text-to-audio | ✗ | ✓ |
| EZAudio (Feng et al., 2024) | General audio | Diffusion Transformer | Text-to-audio | ✗ | ✓ |
| Make-An-Audio (Huang et al., 2023) | General audio | Diffusion | Text-to-audio | ✗ | ✓ |
| Stable Audio Open (Evans et al., 2025) | General audio | Latent diffusion | Text-to-audio | ✗ | ✓ |
| ~~*Respiratory-Specific Systems*~~ *Respiratory-Specific Representation and Synthesis Systems* | | | | | |
| OPERA (Zhang et al., 2024) | Respiratory | Foundation model | Representation learning | ✓ | ✗ |
| **M2D+Resp (Niizumi et al., 2025)** | **General audio + respiratory** | **Masked Modeling Duo (M2D)** | **Representation learning** | ✓ | ✗ |
| RespAgent (Zhang et al., 2026) | Respiratory | Flow matching + agent | Diagnosis + synthesis | ✓ | ✗ |
| *Respiratory Data Augmentation* | | | | | |
| No augmentation (Xia et al., 2022) | Respiratory | - | Training only | ✓ | ✗ |
| Classical augmentation (Xia et al., 2022) | Respiratory | Signal transforms | Data augmentation | ✓ | ✗ |
| GAN-based synthesis (Ramesh et al., 2020) | Respiratory | GAN | Synthetic generation | ✓ | ✗ |
| **LungTTA (ours)** | Respiratory | Diffusion Transformer | ~~Text-to-audio~~ **Text-to-audio generation** | ✓ | ✓ |

## 3 LungTTA

~~LungTTA is a text-to-audio (TTA) framework for synthesizing high-fidelity respiratory sounds, fine-tuned on publicly available recordings to specialize in cough, breathing, and phonation.~~ **LungTTA is a fine-tuned version of Stable Audio Open (Evans et al., 2025), adapted specifically for synthesizing high-fidelity respiratory sounds, including cough, breathing, and phonation.**

~~As illustrated in Figure 1, the pipeline integrates a variational autoencoder (VAE), a T5-based text conditioning module, a transformer-based diffusion backbone with cross-attention and a noise scheduler, and a retrieval memory.~~ **As illustrated in Figure 1, it retains the pre-trained Stable Audio Open variational autoencoder (VAE) for compressing waveforms into latent representations and decoding generated latents back into audio, the T5-based text encoder for producing semantic conditioning embeddings, and the diffusion transformer (DiT) with cross-attention and a noise scheduler for latent-space generation.**

~~The latent diffusion backbone is adapted from Stable Audio Open (Evans et al., 2025), where audio is compressed into a latent representation using a VAE, generated in latent space via a diffusion transformer (DiT), and decoded back into waveform space.~~

~~During training, real respiratory audio is passed through the VAE encoder to guide latent alignment, while inference is performed purely from text and conditioning inputs.~~ **During training, real respiratory recordings are encoded by the VAE and used to fine-tune the diffusion backbone for respiratory acoustics, whereas inference is performed from text and conditioning inputs alone.**

~~On top of this backbone, LungTTA introduces respiratory-specific prompt conditioning, retrieval-guided memory augmentation, and watermarking for traceability.~~ **Building on this pre-trained backbone, LungTTA further introduces respiratory-specific prompt conditioning, retrieval-guided memory augmentation, and watermarking for traceability.**

**Figure 1 was modified to improve text clarity and better illustrate the flow of the pipeline.**

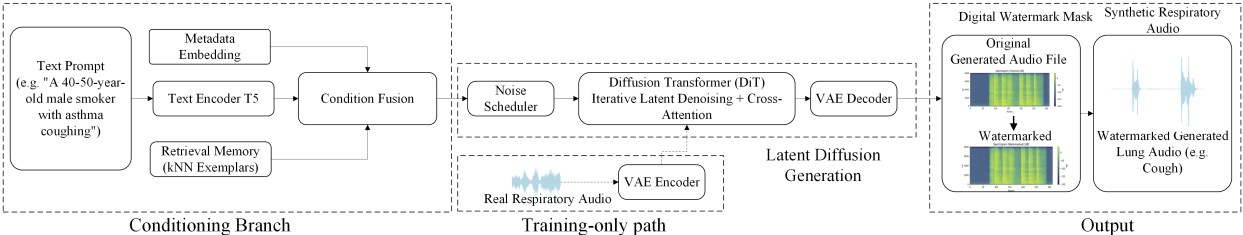

Figure 1: Overview of the proposed LungTTA pipeline. ~~The framework takes textual respiratory prompts as input and generates synthetic lung sounds through a latent diffusion architecture.~~

Text conditioning is provided through a T5-based encoder. During fine-tuning, the pretrained backbone is exposed to prompts derived from dataset metadata, allowing it to shift from general-purpose audio generation toward clinically meaningful respiratory events. The LungTTA generated audio samples are available at `https://lungtta.github.io/audio_samples/`, and the source code ~~will be released upon acceptance~~ **is available at `https://github.com/anonymous-project-code/lungtta_code`.**

## 3.1 Architectural Components

**Backbone and Fine-Tuning** ~~LungTTA builds upon the Stable Audio Open architecture (Evans et al., 2025), which consists of a variational audio autoencoder (156M parameters), a T5-based text encoder (109M parameters), and a diffusion transformer (DiT) with approximately 1.06B parameters operating in latent space.~~ **LungTTA is initialized from the pretrained Stable Audio Open architecture (Evans et al., 2025), including its variational audio autoencoder, T5-base text encoder, and latent diffusion transformer (DiT).**

~~The autoencoder compresses raw waveforms into a 64-channel latent representation at a reduced temporal resolution (∼21.5 Hz), which enables efficient generation of high-resolution audio at 44.1 kHz.~~ **The variational audio autoencoder contains approximately 156M parameters and compresses a waveform sampled at $44.1$ kHz into a 64-channel latent representation with a temporal resolution of approximately $21.5$ Hz.**

**The component labelled "VAE Encoder" in Figure 1 is the encoder of this pretrained variational audio autoencoder; it is not a separate respiratory feature extractor introduced by LungTTA. During training, this encoder maps real respiratory waveforms into the latent space used by the diffusion model. During inference, no reference audio is provided, and the generated latent representation is converted back into waveform space using the corresponding pretrained autoencoder decoder.**

~~Rather than training a model from scratch, we fine-tune this pretrained backbone on respiratory audio, allowing it to retain general audio generation capabilities while adapting to domain-specific acoustic patterns.~~ **The T5-base text encoder contains approximately 109M parameters, while the DiT contains approximately 1.06B parameters and performs text-conditioned denoising in the latent audio space. LungTTA does not use LoRA or adapter-based fine-tuning.**

~~In contrast to the original backbone (~1.3B total parameters), the LungTTA variant used here contains 96.41M *trainable* parameters, making it feasible to train on moderate GPU resources.~~ **The pretrained audio autoencoder and T5-base text encoder are kept frozen, while the 96.41M trainable parameters correspond to the selected diffusion and respiratory-conditioning layers updated during fine-tuning. This parameter count includes only the parameters updated during respiratory-domain fine-tuning and does not represent the total size of the inherited Stable Audio Open architecture.**

~~Training follows a latent diffusion objective, where the model learns to denoise latent representations conditioned on text prompts.~~ **The diffusion objective is computed during training before watermarking. Watermarking is applied only to the final generated waveform as a post-generation traceability operation and is therefore not included in the training-loss computation.**

**Text and Prompt Conditioning** **The following sections and equations were revised and expanded to provide a deeper discussion of the technical contributions.**

Let $p = (p_1, \ldots, p_L)$ denote a tokenized prompt containing $L$ valid tokens. The pretrained T5-base encoder produces a sequence of token-level representations rather than a single pooled vector:

$$H = f_{\text{text}}(p) = [h_1, \ldots, h_L] \in \mathbb{R}^{L \times 768}. \tag{1}$$

Here, $h_j \in \mathbb{R}^{768}$ is the contextual representation of the $j$-th prompt token. The complete sequence $H$ is retained as the primary cross-attention conditioning input to the diffusion transformer. A separate prompt-level query vector is obtained by pooling the valid token representations:

$$q = \text{Pool}(H) \in \mathbb{R}^d. \tag{2}$$

Thus, $q$ is neither the raw prompt $p$ nor an individual prompt token. Instead, it is a fixed-dimensional representation of the complete prompt used to query the retrieval-based memory bank.

**Retrieval-Based Memory Bank** The memory bank is constructed exclusively from the respiratory training corpus used for fine-tuning. It contains $N$ prompt-conditioning embeddings:

$$\mathcal{M} = \{e_i\}_{i=1}^N, \qquad e_i \in \mathbb{R}^d. \tag{3}$$

Each $e_i$ represents the prompt-conditioning information associated with one training example. Given the prompt-level query $q$, the index set containing the $k$ nearest memory entries is defined as

$$I_k(q) = \text{TopK}_{i \in \{1, \ldots, N\}} \left( -d(q, e_i), \, k \right) = \{i_1, \ldots, i_k\}. \tag{4}$$

Here, $d(\cdot, \cdot)$ denotes the retrieval distance function. The variable $k$ denotes the number of memory entries retrieved for each query and is unrelated to the prompt-token sequence length $L$. In the main experiments, $k = 4$. The resulting memory-conditioning representation is defined as

$$Z_{\text{mem}} = [e_{i_1}; e_{i_2}; \ldots; e_{i_k}] \in \mathbb{R}^{k \times d}. \tag{5}$$

Retrieval is therefore performed once for the complete prompt rather than independently for each prompt token. The retrieved embeddings are appended as auxiliary conditioning tokens and do not replace the original T5 token sequence. The procedure does not map a freely written prompt to a single training prompt, retrieve waveform audio, or directly copy a training recording. Instead, the retrieved embeddings provide the diffusion model with conditioning priors associated with similar examples in the respiratory training corpus. These priors guide generation towards respiratory acoustic structures observed during fine-tuning, particularly for comparatively infrequent prompt attributes, while preserving the semantic information contained in the original prompt.

**Metadata Conditioning and Conditioning Fusion** Let $m$ denote the structured metadata available for a recording. Depending on availability in the source dataset, this metadata may include sound type, dataset source, age range, gender, smoking status, respiratory condition, and recording setup. The metadata encoder maps these categorical or numerical attributes into a sequence of conditioning representations:

$$Z_{\text{meta}} = g_{\text{meta}}(m) \in \mathbb{R}^{R \times d_{\text{meta}}}. \tag{6}$$

Here, $R$ is the number of metadata-conditioning tokens and $d_{\text{meta}}$ is their embedding dimension. Because the included datasets do not provide identical metadata fields, unavailable attributes are explicitly represented as missing rather than inferred.

Before conditioning fusion, the prompt, memory, and metadata representations are projected into a common conditioning dimension $d_c$:

$$\widetilde{H} = HW_H \in \mathbb{R}^{L \times d_c}, \qquad \widetilde{Z}_{\text{mem}} = Z_{\text{mem}}W_{\text{mem}} \in \mathbb{R}^{k \times d_c}, \qquad \widetilde{Z}_{\text{meta}} = Z_{\text{meta}}W_{\text{meta}} \in \mathbb{R}^{R \times d_c}. \tag{7}$$

The corresponding projection matrices are

$$W_H \in \mathbb{R}^{768 \times d_c}, \qquad W_{\text{mem}} \in \mathbb{R}^{d \times d_c}, \qquad W_{\text{meta}} \in \mathbb{R}^{d_{\text{meta}} \times d_c}. \tag{8}$$

The complete conditioning sequence is then defined as

$$Z_{\text{cond}} = \left[\widetilde{H}; \widetilde{Z}_{\text{mem}}; \widetilde{Z}_{\text{meta}}\right] \in \mathbb{R}^{(L+k+R) \times d_c}. \tag{9}$$

The bracket notation denotes concatenation along the conditioning-token dimension after the three representations have been projected into the same feature space. The DiT attends to this combined sequence during latent denoising.

**Prompt Engineering** ~~The prompts are constructed directly from dataset metadata rather than written manually.~~ **The training prompts are constructed from metadata associated with the respiratory recordings in the training corpus rather than being manually written for each sample.**

~~Each one follows a simple template, for example:~~ *~~"A person is [activity], recorded in [condition]."~~* **Each prompt follows a consistent template, such as** *"A person is [activity], recorded in [condition]."*

~~The activity specifies the sound type, such as coughing or breathing, while the condition includes attributes like age, gender, smoking status, or recording setup.~~ **The activity specifies the respiratory sound type, such as coughing, breathing, or phonation, while the condition incorporates available attributes such as age range, gender, smoking status, respiratory condition, or recording setup.**

~~This keeps the prompts consistent with the underlying data.~~

~~For instance, we use prompts such as~~ `"A person is vocalizing the sustained vowel sound /a/."` ~~or~~ `"A person is coughing, male, aged 40-50, smoker, asthmatic."`~~.~~ **Example prompts include** `"A person is vocalizing the sustained vowel sound /a/."` **and** `"A person is coughing, male, aged 40-50, smoker, asthmatic.".`

~~Similar strategies have been used in AudioLDM2 Liu et al. (2023) and RespAgent Zhang et al. (2026), where text descriptions are derived from metadata rather than free-form annotation.~~ **Similar metadata-derived prompt strategies have been used in AudioLDM2 (Liu et al., 2023) and RespAgent (Zhang et al., 2026).**

**At inference time, the input prompt is encoded directly by the T5-base encoder, while the memory bank supplies additional conditioning embeddings retrieved using the prompt-level query representation. The model is therefore guided primarily by the input prompt while also receiving auxiliary information derived from similar examples in the respiratory training corpus.**

**Ethical Considerations and Traceability** Generated respiratory recordings may resemble real audio and could be mistaken for genuine recordings if their synthetic origin is not clearly indicated. LungTTA therefore applies a low-amplitude watermark only after the diffusion model has generated the latent audio representation and the pretrained autoencoder decoder has reconstructed the waveform. Let $x_{\text{gen}}(t)$ denote the unwatermarked generated waveform. The watermarked waveform is defined as

$$x_{\text{wm}}(t) = x_{\text{gen}}(t) + \lambda w(t). \tag{10}$$

Here, $w(t)$ is the watermark signal and $\lambda \ll 1$ controls its amplitude. The training objective is computed before this watermarking operation. Similarly, primary audio-quality evaluations use the unwatermarked output unless the effect or robustness of watermarking is being evaluated explicitly. The watermark provides a traceability mechanism but does not guarantee resistance to removal or establish the clinical validity of the generated recordings. Since LungTTA is fine-tuned on public respiratory datasets, its outputs may also reflect demographic, clinical, and recording-condition imbalances in those sources. Generated audio is therefore intended for research and data-augmentation purposes and should not be treated as diagnostic evidence.

### 3.2 PRISM: Pulmonary Respiratory Integrity & Similarity Metric

Conventional metrics such as FAD, KL divergence, and Inception Score ~~measure~~ **primarily describe** global distributional similarity between real and generated audio.

~~However, respiratory sounds are often interpreted at the event level, where clinical relevance depends on temporal evolution, breathing-cycle structure, tonal wheeze-like content, transient crackle-like behavior, and the distribution of energy across respiratory frequency bands.~~ **Respiratory recordings, however, are often assessed at the event level, where relevant structure includes temporal evolution, breathing-cycle dynamics, tonal wheeze-like content, transient crackle-like events, and spectral-energy distribution across respiratory frequency bands.**

~~To address this limitation, we introduce~~ **PRISM (Pulmonary Respiratory Integrity & Similarity Metric)**, ~~a domain-aware similarity metric defined for a real waveform $x$ and a generated waveform $\hat{x}$ as~~ **To capture these properties, we introduce PRISM (Pulmonary Respiratory Integrity & Similarity Metric), a signal-derived respiratory-structure similarity metric between a reference recording $x$ and a generated recording $\hat{x}$:**

$$\text{PRISM}(x, \hat{x}) = \sum_{i=1}^{5} w_i S_i(x, \hat{x}), \tag{11}$$

where $S_i$ are component-level respiratory similarity terms and $w_i \geq 0$ are non-negative weights.

In our implementation, PRISM is defined as

$$\text{PRISM}(x, \hat{x}) = 0.30\, S_{\text{traj}} + 0.20\, S_{\text{cycle}} + 0.20\, S_{\text{wheeze}} + 0.15\, S_{\text{crackle}} + 0.15\, S_{\text{band}}. \tag{12}$$

~~These components are selected to capture complementary aspects of respiratory sound structure, including temporal evolution, coarse breathing dynamics, tonal coherence, transient events, and frequency energy distribution, which are commonly used in respiratory sound analysis (Pramono et al., 2017).~~

**Here, $S_{\text{traj}}$ measures MFCC-based spectro-temporal trajectory similarity after dynamic time warping alignment; $S_{\text{cycle}}$ compares RMS-envelope structure and coarse respiratory-cycle balance; $S_{\text{wheeze}}$ measures band-limited phase coherence in the 400–1600 Hz range as an acoustic proxy for wheeze-like tonal structure; $S_{\text{crackle}}$ compares frame-wise kurtosis as a proxy for short transient crackle-like structure; and $S_{\text{band}}$ measures agreement in spectral-energy distribution across predefined respiratory frequency bands.**

~~Here, $S_{\text{traj}}$ measures trajectory similarity, defined as the agreement in the temporal evolution of spectral features over time using Mel-frequency cepstral coefficients (MFCC) aligned via dynamic time warping (DTW), $S_{\text{cycle}}$ measures similarity of the root mean square (RMS) energy envelope and coarse respiratory cycle balance, $S_{\text{wheeze}}$ measures band-limited phase coherence in the 400-1600 Hz range, $S_{\text{crackle}}$ measures similarity of frame-wise kurtosis as a proxy for transient crackle-like structure, and $S_{\text{band}}$ measures band-energy similarity, defined as the agreement in the distribution of spectral energy across predefined respiratory frequency bands.~~

~~Higher PRISM values indicate stronger agreement between generated and real respiratory structure.~~ **Higher PRISM values indicate stronger agreement between generated and reference respiratory structure.**

**The five components are selected to capture complementary respiratory-signal properties rather than to provide a clinical diagnosis. MFCC and cepstral features are commonly used in lung-sound analysis because they compactly represent spectral-envelope information, while DTW provides a way to compare trajectories that may differ in onset, duration, or local timing (Sengupta et al., 2016; Cuturi & Blondel, 2017). The envelope-based cycle term reflects the temporal organisation of respiratory recordings around inspiratory and expiratory phases (Jácome et al., 2019). The wheeze-like and crackle-like terms are motivated by standard respiratory-sound descriptions of wheezes as continuous tonal events and crackles as brief discontinuous adventitious sounds (Bohadana et al., 2014; Grønnesby et al., 2017). The band-energy term provides a compact summary of respiratory-relevant spectral-energy distribution.**

**PRISM Weight Study** **The following section has been added to present the PRISM weight study.** The weights in Eq. 12 are transparent heuristic choices intended to balance five complementary components rather than clinically optimised parameters. The trajectory component, $S_{\text{traj}}$, receives the highest weight because it captures global spectro-temporal similarity; $S_{\text{cycle}}$ and $S_{\text{wheeze}}$ receive intermediate weights because they represent respiratory envelope structure and tonal coherence; and $S_{\text{crackle}}$ and $S_{\text{band}}$ receive smaller weights because they capture narrower transient and spectral-energy characteristics. To assess sensitivity to these choices, we constructed 250 controlled comparisons from 50 reference recordings spanning identical, mild, moderate, strong, and mismatched pairs, with the expected ordering identical > mild > moderate > strong > mismatched. Candidate non-negative weights summing to one were evaluated using a grid step of 0.05, producing 10,626 schemes, including a balanced subset of 3,876 schemes in which each component received a weight of at least 0.05. Performance was assessed using monotonic ordering, adjacent validation gaps, Spearman correlation, MSE to the ordinal target, two AUC measures, and weight entropy. As shown in Figure D.1, the proposed, equal, and empirically selected weights all preserve the expected ordering, although the empirical schemes place greater emphasis on band-energy similarity. The proposed weights achieve a minimum adjacent gap of 0.059, a Spearman correlation of 0.797, and a same/perturbed-versus-mismatched AUC of 0.874, while 99.02% of all tested schemes preserve monotonic behaviour. These results indicate that PRISM is not strongly dependent on the selected weighting, supporting retention of the proposed heuristic weights rather than post hoc optimisation for LungTTA performance.

**PRISM Metric Validation on Ground-Truth Data** **This section was modified to extend the PRISM validation from 20 to 50 recordings.** We validate PRISM using 50 held-out ground-truth cough recordings under identical, perturbed, and mismatched conditions. Identical pairs compare each

recording with itself, while perturbed pairs compare it with transformed versions of the same waveform using amplitude scaling of 0.70, a random zero-padded temporal shift within $\pm 50$ ms, time stretching at a rate of 1.05, downsampling to 12 kHz followed by resampling to 22.05 kHz, sixth-order low-pass filtering at 4 kHz, and additive Gaussian noise at 20 dB signal-to-noise ratio. Five independently seeded perturbations are averaged per recording. For mismatched pairs, each recording is compared with a randomly selected different ground-truth recording across ten independently seeded assignments, which are also averaged per source recording; therefore, the final statistical sample size remains $N = 50$. PRISM produces the expected ordering, with scores of $1.000 \pm 0.000$ for identical pairs, $0.306 \pm 0.143$ for perturbed pairs, and $0.179 \pm 0.042$ for mismatched pairs, with bootstrap 95% confidence intervals of $[1.000, 1.000]$, $[0.268, 0.346]$, and $[0.167, 0.190]$, respectively. Perturbed pairs score significantly higher than mismatched pairs, with a mean paired difference of 0.127, a bootstrap 95% confidence interval of $[0.096, 0.160]$, Holm-adjusted $p < 0.001$, and Cohen's $d_z = 1.08$. As shown in Figure 2, PRISM assigns maximum similarity to identical recordings, intermediate similarity to transformed versions of the same recording, and lower similarity to unrelated recordings, while the component-level analysis shows that perturbed recordings retain greater cycle, transient, and band-energy similarity than mismatched recordings. Additional results using 20 recordings are provided in Appendix C.

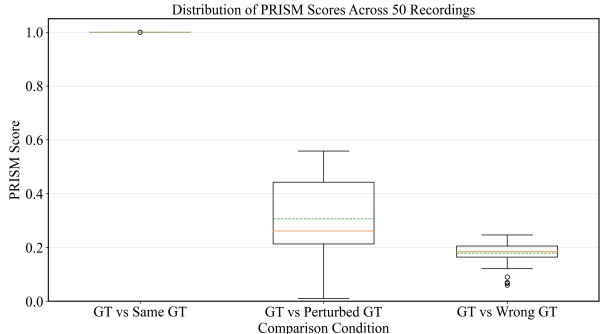

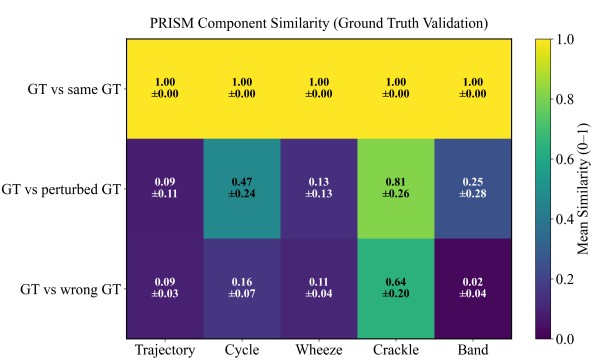

(a) Overall PRISM scores for identical, perturbed, and mismatched ground-truth pairings.

(b) Component-level PRISM similarity, reported as mean $\pm$ standard deviation across 50 recordings.

Figure 2: Controlled PRISM validation on 50 held-out ground-truth cough recordings. Left: identical pairs receive the highest scores, perturbed versions of the same recordings receive intermediate scores, and mismatched recordings receive the lowest scores. Right: component-level similarities for trajectory, cycle, wheeze-like coherence, crackle-like transient structure, and respiratory band-energy terms.

## 4 Experimental Setup

### New Section title

**Dataset**  We train and evaluate LungTTA using nine publicly available respiratory audio datasets, comprising a total of 116,660 recordings covering coughs, breathing patterns, and vowel phonation. Table 2 provides an overview of the datasets, including recording devices and sample counts. As the datasets originate from different sources, a preprocessing step was required to standardize them. All audio was resampled, corrupted files were removed, and filenames were normalized into a consistent format. Metadata was retained and converted into a unified JSON representation to support prompt conditioning and downstream evaluation. The data was split into training (80%), validation (10%), and testing (10%) sets. To support conditioning and analysis, recordings were grouped into 13 categories based on their original labels and recording protocols, including cough, breathing, vowel phonation, counting tasks, and stethoscope-based recordings. Where available, finer-grained labels such as *deep* and *shallow* breathing, or variations in cough intensity (e.g., *heavy* or *shallow*), were retained from the source annotations.

Table 2: Summary of datasets and number of recordings.

| Dataset | Device | # Recordings. |
|---|---|---|
| COVID-19 Sounds (Han et al., 2022) | Microphone | 53,449 |
| UK COVID-19 (Coppock et al., 2024) | Microphone | 25,706 |
| COUGHVID (Orlandic et al., 2021) | Microphone | 20,072 |
| CoronaHack (Thandu & Gera, 2024) | Microphone | 1,400 |
| MMLung (Mosuily et al., 2023) | Microphone | 560 |
| Vowels (David Andres Rubiano Venegas, 2019) | Microphone | 1,676 |
| HF Lung (Hsu et al., 2022) | Stethoscope | 9,765 |
| Respiratory TR (Altan et al., 2017) | Stethoscope | 3,696 |
| KAUH Lung (Fraiwan et al., 2021) | Stethoscope | 336 |

**Training** LungTTA is trained as a text-conditioned latent diffusion model for synthetic respiratory audio generation on a high-performance computing node equipped with a single NVIDIA A100 GPU, 48 CPU cores and 180 GB RAM, using PyTorch and Librosa (McFee et al., 2015). The dataset contains 116,660 recordings, split into 93,328 training, 11,666 validation, and 11,666 test samples.

**Because the source recordings vary in sampling rate, channel configuration, and duration, all waveforms were standardised before batching. Audio was resampled to 44.1 kHz and represented as stereo input to match the Stable Audio Open backbone, with mono recordings duplicated across the two channels. Longer recordings were divided into fixed-length segments of 262,144 samples, corresponding to approximately 5.94 s, while shorter recordings were zero-padded to the same length.**

Text prompts are encoded using a T5 encoder, while audio is mapped into a latent space using a pretrained VAE that remains frozen during training. The generative backbone is a diffusion transformer (DiT) with 24 layers, 1536 embedding dimension and 24 attention heads, operating on 64-channel latent representations. In addition to text conditioning, temporal variables (start time and total duration) are included as global conditioning signals. A retrieval-based memory module is used during training, where the top-$k$ nearest embeddings ($k = 4$) from a precomputed memory bank are incorporated as auxiliary conditioning, where $k = 4$ balances diversity and conditioning stability based on empirical observations and prior retrieval-augmented methods (Lewis et al., 2020). The model is optimized using AdamW with a learning rate of $5 \times 10^{-5}$ and weight decay $1 \times 10^{-3}$, and trained for 250,000 steps with checkpoints saved every 5,000 steps and validation performed at each epoch. Classifier-free guidance is applied during sampling with guidance scales between 4 and 7.

~~The full model contains approximately 96.41 M parameters and requires around 30 hours of training.~~ **Approximately 96.41 M parameters are updated during fine-tuning and requires around 30 hours of training.**

## 5 Results

We evaluate LungTTA using a mix of objective metrics, listening studies, ablation experiments, and a downstream classification task. The aim is to understand both the quality of the generated audio and how well it follows the conditioning prompts, as well as whether the synthetic data is useful in practice. A qualitative example is shown in Figure A.1, where LungTTA produces spectrograms that better match the timing and energy patterns of real cough signals compared to baseline models. The following sections report quantitative results, prompt diversity experiments, comparisons with existing methods, ablations, and downstream performance.

### 5.1 Objective Evaluation

We evaluate generated lung sounds using Fréchet Audio Distance (FAD), Kullback–Leibler divergence (KL), Inception Score (IS), and PRISM. Lower values indicate better performance for FAD and KL, while higher values are preferred for IS and PRISM. Further definitions and implementation details for these evaluation metrics are provided in Appendix A. To study the effect of prompt diversity, we test 1, 5, 10, 20, and 50 conditioning prompts under the same setup.

We evaluate generated lung sounds using Fréchet Audio Distance (FAD), Kullback-Leibler divergence (KL), Inception Score (IS), and PRISM. Lower values indicate better performance for FAD and KL, while higher values are preferred for IS and PRISM. Further definitions for these evaluation metrics are provided in Appendix A. To study the effect of prompt diversity, we test 1, 5, 10, 20, and 50 conditioning prompts under the same setup. As shown in Figure 3, FAD drops from 17.33 (1 prompt) to 1.81 (50 prompts), and KL from 1.94 to 0.19, while IS increases from 1.12 to 1.33. PRISM stays within a narrower range, between 0.17 and 0.21. With fewer than 5 prompts, the results vary more, likely due to limited coverage of the data distribution. Increasing the number of prompts improves diversity and leads to more stable outputs. After around 20 prompts, the gains become smaller, suggesting that most of the dominant variations are already captured. Based on this, we use 20 prompts in the remaining experiments.

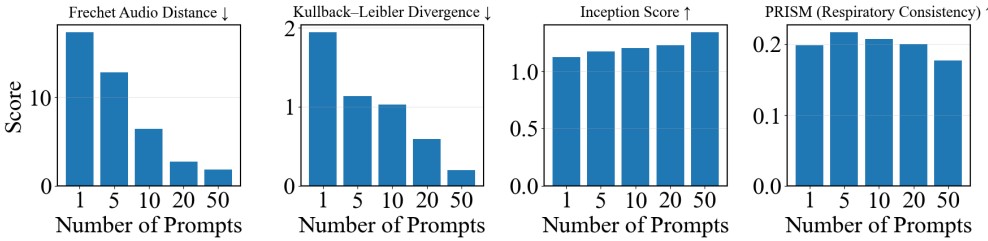

Figure 3: Effect of prompt diversity on LungTTA performance.

**Comparison with State-of-the-Art Methods**   All models are evaluated under identical conditions. Table 3 shows that LungTTA achieves the lowest FAD (2.72) and KL (0.50). Make-An-Audio achieves the highest IS (1.54), while RespAgent obtains the highest PRISM (0.24). This is expected, as RespAgent conditions on both text and reference audio. LungTTA, which is text-only, achieves a comparable PRISM score of 0.23.

Table 3: Comparison with state-of-the-art text-to-audio models.

| Model | FAD ↓ | KL ↓ | IS ↑ | PRISM ↑ |
|---|---|---|---|---|
| AudioLDM2 (Liu et al., 2023) | 6.21 | 0.71 | 1.35 | 0.15 |
| EzAudio (Feng et al., 2024) | 6.59 | 0.52 | 1.46 | 0.14 |
| Make-An-Audio (Huang et al., 2023) | 15.95 | 1.34 | **1.54** | 0.14 |
| StableAudioOpen (Evans et al., 2025) | 6.73 | 0.67 | 1.39 | 0.21 |
| RespAgent (Zhang et al., 2026) | 8.53 | 0.53 | 1.44 | **0.24** |
| **LungTTA (Ours)** | **2.72** | **0.50** | 1.22 | 0.23 |

**Ablation Study**   ~~We analyze the contribution of the memory bank and watermarking using FAD, KL, IS, and PRISM (Table 4).~~ **The ablation results indicate a trade-off rather than uniform improvement across all metrics.**

~~The baseline model achieves an FAD of 6.73 and a PRISM score of 0.21. Introducing the memory bank leads to a substantial improvement in FAD (6.73 to 2.72), while IS decreases from 1.39 to 1.23.~~ **The memory bank reduces FAD from 6.73 to 2.72, suggesting closer alignment with the real respiratory-audio distribution, but decreases IS from 1.39 to 1.23 and PRISM from 0.21 to 0.18 in the**

memory-only setting. **This may reflect stronger retrieval guidance that improves distributional similarity while reducing sample diversity or increasing retrieval bias.**

~~Adding watermarking produces minimal change in FAD and IS, although KL decreases from 0.67 to 0.47. IS remains largely unchanged (1.39 to 1.38), indicating that watermarking does not affect generation quality.~~ **Watermarking changes FAD and IS only marginally, indicating limited influence on generation quality. Although watermarking alone reduces KL from 0.67 to 0.47, this result is treated as an incidental distributional effect and not as evidence that watermarking improves perceptual quality.**

~~The full LungTTA model achieves a KL of 0.50, while the lowest KL (0.47) is observed when watermarking is applied alone. The full model also achieves the highest PRISM score (0.23).~~ **The full LungTTA model achieves the highest PRISM score of 0.23, suggesting that the combined configuration provides the strongest respiratory-structure agreement among the ablated settings.**

~~Overall, the memory component provides the main improvement in FAD, while watermarking has limited effect on FAD and IS but reduces KL.~~ **Additional experiments are provided in Appendix B.**

Table 4: Ablation study evaluating the impact of the proposed memory and watermarking modules.

| Setting | Memory Bank | Watermarking | FAD ↓ | KL ↓ | IS ↑ | PRISM ↑ |
|---|---|---|---|---|---|---|
| Baseline Model **(Stable Audio Open)** | × | × | 6.73 | 0.67 | **1.39** | 0.21 |
| Baseline + Watermarking | × | ✓ | 6.74 | **0.47** | 1.38 | 0.20 |
| Baseline + Memory Bank | ✓ | × | 2.72 | 0.59 | 1.23 | 0.18 |
| **LungTTA (Full Model)** | ✓ | ✓ | **2.72** | 0.50 | 1.22 | **0.23** |

**Watermark Detection and Robustness** **New Section** We evaluated the watermark using 50 LungTTA-generated recordings. Fifteen recordings were used to select the detection threshold, and the remaining 35 were used for testing. Each test recording was evaluated before and after watermarking, and the watermarked versions were also tested after amplitude scaling, additive noise, resampling, and MP3 compression. The watermark was detected in 31 of 35 unmodified recordings, giving a detection rate of 88.57%, while none of the non-watermarked recordings was incorrectly detected. Overall accuracy was 94.29%, with an F1 score of 0.939 and an AUC of 1.000. As shown in Figure F.1, detection remained at 88.57% after amplitude scaling and resampling to 22.05 kHz, decreased to 82.86% after MP3 compression at 64 kbps, and to 77.14% after adding noise at 10 dB SNR (Signal-to-Noise Ratio). The waveform correlation before and after watermarking was 0.999842, showing that the watermark caused only a small change to the signal. Detection failed after resampling to 16 kHz and low-pass filtering at 8 kHz, so the watermark is intended for basic traceability rather than secure protection.

**Downstream Evaluation** To assess the usefulness of the generated data for downstream tasks, we follow a realistic evaluation protocol inspired by prior work on audio-based COVID-19 detection (Han et al., 2022). In particular, we adopt a participant-independent data split and consistent training and evaluation settings to avoid over-optimistic performance estimates. For the baseline methods, we use a VGGish-based architecture, where VGGish, a VGG-like Convolutional neural network pretrained on AudioSet, serves as the feature extraction backbone, followed by pooling and fully connected layers for classification. For LungTTA, we additionally evaluate performance using an Audio Spectrogram Transformer (AST)-based model (Gong et al., 2021), where mid-level representations are extracted from the transformer backbone and used for classification. All models are trained and evaluated on identical data splits, and the decision threshold is selected based on validation performance before testing. For training, the real dataset is imbalanced, consisting of 7,660 positive and 14,188 negative samples. To mitigate this imbalance and improve generalization, we augment the positive class with 7,000 synthetic samples generated by LungTTA, resulting in a more balanced training set with 14,660 positive and 14,188 negative samples. As shown in Table 5, LungTTA consistently improves downstream classification performance over classical and GAN-based augmentation. The VGGish-based model achieves an F1 score of 0.6213, compared to 0.6087 with classical augmentation

and 0.5801 without augmentation. The AST-based configuration achieves an AUC of 0.7791 and sensitivity of 0.7027, compared to 0.7701 AUC and 0.6995 sensitivity for the VGGish-based LungTTA model.

Table 5: Downstream COVID-19 cough classification performance under identical evaluation settings.

| Method | Model | F1 ↑ | AUC ↑ | Sensitivity ↑ | Specificity ↑ |
|---|---|---|---|---|---|
| No Augmentation | VGGish-based | 0.5801 | 0.7331 | 0.6985 | 0.6397 |
| **No Augmentation** | **AST (mid-layer)** | **0.6059** | **0.7694** | **0.6375** | **0.7629** |
| Classic Augmentation | VGGish-based | 0.6087 | 0.7631 | 0.6672 | 0.7333 |
| **Classic Augmentation** | **AST (mid-layer)** | **0.5979** | **0.7624** | **0.6757** | **0.7032** |
| GAN | VGGish-based | 0.5029 | 0.7540 | 0.4123 | **0.8846** |
| **GAN** | **AST (mid-layer)** | **0.5891** | **0.7730** | **0.5501** | **0.8388** |
| **LungTTA (Ours)** | VGGish-based | **0.6213** | 0.7701 | 0.6995 | 0.7196 |
| **LungTTA (Ours)** | AST (mid-layer) | 0.6170 | **0.7791** | **0.7027** | 0.7080 |

## 5.2 Asthma Classification with Augmented Training Data

**New Section** We evaluated asthma classification using a VGGish-based binary classifier. The input for all experiments was the three-cough recording. The training set was balanced with 5,884 asthma-negative samples and 5,884 asthma-positive samples. The validation and test sets were kept unchanged and contained only real recordings. The validation set included 5,459 samples, and the test set included 5,587 samples.

For the no-augmentation baseline, real asthma-positive samples were repeated to balance the training set. For offline classical augmentation, 2,942 asthma-positive recordings were generated from the real positive training samples using waveform transformations and saved before training. For LungTTA, 2,942 synthetic asthma-positive cough recordings were generated and saved before training.

Table 6: Asthma classification performance using VGGish.

| Method | Accuracy ↑ | F1 ↑ | AUC ↑ | Sensitivity ↑ | Specificity ↑ |
|---|---|---|---|---|---|
| No Augmentation | 0.5767 | 0.2066 | 0.5506 | 0.4695 | 0.5910 |
| GAN Augmentation | 0.5532 | 0.2076 | 0.5517 | **0.4985** | 0.5605 |
| Classical Augmentation | 0.5948 | 0.2161 | 0.5658 | 0.4756 | 0.6106 |
| **LungTTA (Ours)** | **0.6263** | **0.2226** | **0.5852** | 0.4558 | **0.6490** |

LungTTA achieved the highest accuracy, F1 score, AUC, specificity, and balanced accuracy. Compared with offline classical augmentation, LungTTA improved AUC from 0.5658 to 0.5852 and accuracy from 0.5948 to 0.6263.

## 5.3 Subjective Evaluation

The subjective evaluation is blinded. Recordings from the evaluated models were anonymised and presented in randomised order, model identities were concealed, and all participants rated the samples independently without access to the scores of others. In the general-listener study, 32 participants each evaluated six recordings per model, yielding 186 ratings per model, using five-point Likert scales for Overall Quality (OVL) and Relevance to Text (REL) mapped to a 0–100 range. Ground-truth recordings achieved mean scores of 87.23 for OVL and 79.03 for REL, while LungTTA achieved 80.91 and 75.13, respectively; RespAgent achieved 59.27 and 58.97, AudioLDM2 achieved 58.60 and 53.09, Stable Audio Open achieved 56.18 and 55.65, and EZAudio achieved 55.24 and 52.69. For the expert evaluation, three respiratory-health professionals independently assessed the same six recordings per synthetic model, yielding 18 ratings per model, under the

same blinded and randomised conditions. In addition to OVL and REL, the experts rated Clinical Relevance for Assessment (CRA), which was developed in consultation with respiratory-health professionals to assess whether a recording contains respiratory patterns that may support clinical interpretation; the task did not involve diagnosis, verification of disease labels, or confirmation of clinically valid disease signatures. LungTTA achieved mean expert scores of 58.33 for OVL, 44.44 for REL, and 38.89 for CRA, compared with RespAgent at 56.94, 43.06, and 36.11; AudioLDM2 at 48.61, 34.72, and 29.17; Stable Audio Open at 55.56, 40.28, and 33.33; and EZAudio at 36.11, 29.17, and 33.33, respectively. **Inter-rater reliability was assessed using a two-way random-effects, absolute-agreement intraclass correlation coefficient (ICC), with ICC(2,$k$) used as the primary measure because the reported expert scores were averaged across three raters and ICC(2,1) used to describe individual-rater reliability. Bootstrap 95% confidence intervals were calculated from 5,000 recording-level resamples. Average-rater agreement was moderate for OVL, with ICC(2,$k$) = 0.616 and a bootstrap 95% confidence interval of [0.247, 0.784], but poor for REL, with ICC(2,$k$) = 0.248 [–0.493, 0.570], and CRA, with ICC(2,$k$) = 0.220 [–0.487, 0.548]; the corresponding ICC(2,1) values were 0.349, 0.099, and 0.086. A sample from the survey is provided in Appendix H.**

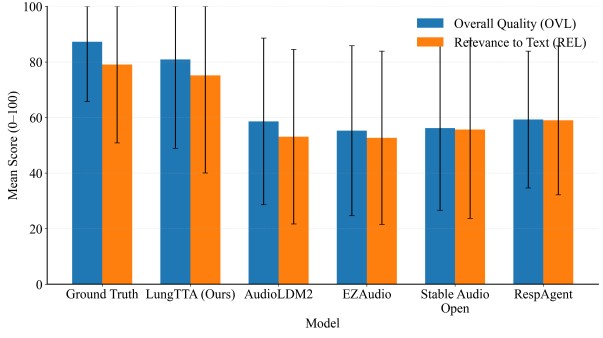

(a) General public evaluation results ($N = 32$). Error bars indicate standard deviation.

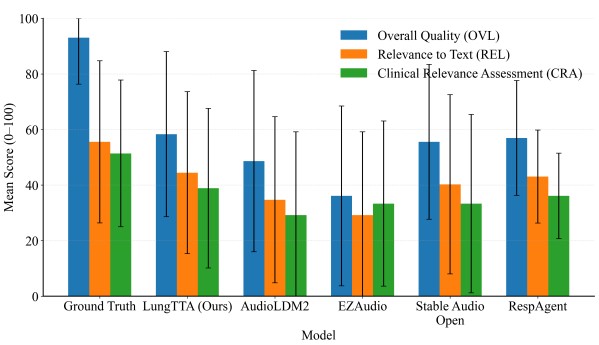

(b) Expert evaluation across OVL, REL, and CRA (0-100 scale, mean ± standard deviation). Scores reflect ratings from three respiratory-health professionals.

Figure 4: Subjective evaluation of generated respiratory audio. Left, ratings from 32 general listeners for Overall Quality (OVL) and Relevance to Text (REL). Right, ratings from three respiratory-health professionals for OVL, REL, and Clinical Relevance for Assessment (CRA). Scores are shown on a 0-100 scale, where 100 is the highest possible score.

## 6 Discussion

The results show that LungTTA improves how closely generated audio matches real respiratory recordings, without being the best on every metric. It achieves the lowest FAD and KL (2.72 and 0.50), compared with AudioLDM2 (6.21, 0.71), EZAudio (6.59, 0.52), Stable Audio Open (6.73, 0.67), and RespAgent (8.53, 0.53), suggesting better distributional alignment. It does not reach the highest IS (1.22 vs 1.54 for Make-An-Audio) and remains slightly below RespAgent on PRISM (0.23 vs 0.24), likely because RespAgent uses reference audio while LungTTA is text-only. The ablation study shows that most of the FAD improvement comes from the memory component (6.73 to 2.72), while the full model achieves the highest PRISM (0.23) and watermarking has little effect on perceptual quality. The listening study supports this, with LungTTA achieving 80.91 (OVL) and 75.13 (REL), compared with around 55–59 for other models, and similar improvements observed in expert-rated OVL, REL, and CRA. In the downstream task, LungTTA improves AUC from 0.7331 (no augmentation) and 0.7631 (classical augmentation) to 0.7701, with the highest AUC of 0.7791 achieved using AST. Overall, these results indicate that LungTTA generates useful task-relevant samples.

# 7 Conclusion

This work introduced LungTTA, a prompt-based text-to-audio framework for generating respiratory sounds using latent diffusion models. LungTTA enables controllable synthesis of cough, breathing, and phonation signals from structured prompts, while incorporating retrieval-based memory guidance and watermarking for traceability. The results show that LungTTA improves the realism and usefulness of synthetic respiratory audio compared to existing baselines, and can support downstream learning in data-scarce settings. LungTTA is not intended to replace real data collection, but to act as a complementary tool for augmenting limited datasets and supporting model development. By enabling scalable and reproducible generation of respiratory audio, it provides a practical pathway for improving robustness and coverage in respiratory health applications.

~~Future work will extend evaluation across more diverse clinical conditions, recording environments, and downstream tasks, and further improve prompt fidelity and clinically relevant structure in generated audio.~~

**Future work will extend evaluation across more diverse clinical conditions, recording environments, and downstream tasks, further improve prompt fidelity and clinically relevant structure in generated audio, conduct a $k$-sensitivity experiment for the retrieval-based memory module, and extend PRISM validation across larger and more diverse respiratory datasets.**

# 8 Compliance with Ethical Standards

This work uses publicly available respiratory sound datasets in line with their licenses and data sharing agreements. All human data collection was approved by relevant ethics committees and conducted according to standard ethical and data protection guidelines. Written consent was obtained from all participants.

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

# Appendix

This appendix provides additional technical and experimental details supporting the main paper. It includes formal definitions of the objective and subjective evaluation metrics, details of the expert clinical assessment protocol, and the downstream classification setup. We also report exploratory architectural variants results to contextualize the final design choices. In addition, the appendix contains the conditioning prompts used for generation, representative prompt variations, and a reconstruction of the survey interface used in the listening study for both general participants and clinical experts.

## A    Evaluation Metrics

We evaluate LungTTA using a combination of objective statistical metrics and subjective perceptual assessments in order to capture realism, distributional similarity, and clinical usefulness in the generated respiratory sounds. All objective metrics are computed within a unified evaluation pipeline in which feature embeddings are extracted from generated samples and their corresponding real recordings before metric computation.

**Figure moved to appendix**

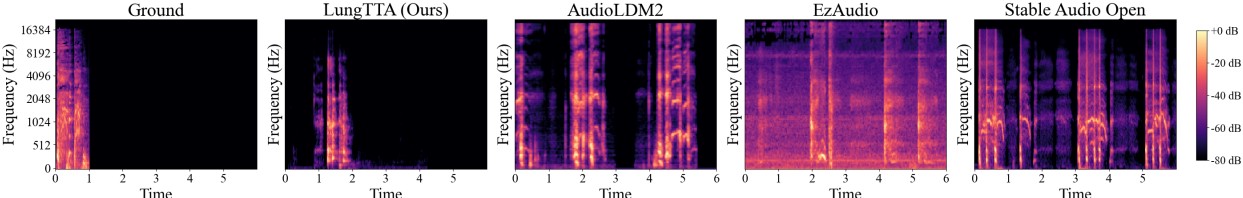

Figure A.1: Spectrogram graph for the prompt *"a cough from a 48-year-old male smoker with no asthma and no COPD."* The panels show the ground-truth recording and outputs generated by LungTTA (Ours), AudioLDM2, EzAudio, and Stable Audio Open.

### A.1    Objective Metrics

#### A.1.1    Fréchet Audio Distance (FAD)

Fréchet Audio Distance quantifies statistical similarity between real and synthetic audio distributions in the embedding space induced by the `FrechetAudioDistance` backend. This representation is intended to retain perceptually relevant attributes such as spectral texture and temporal structure, making FAD a useful proxy for perceptual realism in respiratory audio. Let $\boldsymbol{\mu}_r$ and $\boldsymbol{\mu}_g$ denote the mean feature embeddings of real and generated samples, and let $\boldsymbol{\Sigma}_r$ and $\boldsymbol{\Sigma}_g$ denote their covariance matrices. FAD is defined as

$$\text{FAD} = \|\boldsymbol{\mu}_r - \boldsymbol{\mu}_g\|_2^2 + \text{Tr}\left(\boldsymbol{\Sigma}_r + \boldsymbol{\Sigma}_g - 2\sqrt{\boldsymbol{\Sigma}_r \boldsymbol{\Sigma}_g}\right). \tag{A.1}$$

Lower values indicate that synthetic sounds align more closely with the perceptual distribution of real lung recordings, with zero representing a perfect match.

#### A.1.2    Kullback–Leibler Divergence (KL)

To assess alignment between the probability distributions of real and generated embeddings, we compute the Kullback–Leibler divergence in both sigmoid and softmax activation domains, as returned by the evaluation pipeline. KL divergence measures how much the generated distribution deviates from the real reference distribution and is expressed as

$$D_{\text{KL}}(P\|Q) = \sum_i P(i) \log \frac{P(i)}{Q(i)}. \tag{A.2}$$

Lower values indicate closer distributional agreement, with values approaching zero representing near-identical distributions.

### A.1.3 Inception Score (IS)

We also report Inception Score to evaluate the joint quality and diversity of generated audio. IS is computed from classifier outputs over multiple splits of the synthetic dataset and is defined as

$$\text{IS} = \exp\left(\mathbb{E}_x\, D_{\text{KL}}\left(p(y|x) \,\|\, p(y)\right)\right), \tag{A.3}$$

where $p(y|x)$ is the conditional class distribution for audio sample $x$, and $p(y)$ is the marginal distribution across samples. Higher scores indicate more confident and diverse generations.

### A.2 Subjective Evaluation

Participants rated each audio clip on Overall Quality (OVL) and Relevance to Text (REL) using a 0–100 scale. For each clip $n$, scores are averaged across $K$ participants,

$$\text{OVL}_n = \frac{1}{K}\sum_{k=1}^{K} o_{n,k}, \qquad \text{REL}_n = \frac{1}{K}\sum_{k=1}^{K} r_{n,k}. \tag{A.4}$$

### A.3 Expert Clinical Assessment

An expert evaluation was conducted with three respiratory-health professionals. Each clip was rated on Overall Quality (OVL), Relevance to Text (REL), and Clinical Relevance for Assessment (CRA). For clip $n$, the CRA score is computed as

$$\text{CRA}_n = \frac{1}{K}\sum_{k=1}^{K} c_{n,k}, \tag{A.5}$$

where $K = 3$. Model-level scores are obtained by averaging across all evaluated clips,

$$\text{CRA}_{\text{model}} = \frac{1}{KN}\sum_{n=1}^{N}\sum_{k=1}^{K} c_{n,k}. \tag{A.6}$$

### A.4 Downstream Evaluation

We report standard classification metrics. For completeness, the definitions are given below.

**F1 Score**

$$\text{F1} = \frac{2 \cdot TP}{2 \cdot TP + FP + FN} \tag{A.7}$$

**Sensitivity**

$$\text{Sensitivity} = \frac{TP}{TP + FN} \tag{A.8}$$

**Specificity**

$$\text{Specificity} = \frac{TN}{TN + FP} \tag{A.9}$$

**AUC**

$$\text{AUC} = \int_0^1 \text{TPR}(t)\, d(\text{FPR}(t)) \tag{A.10}$$

# B Exploratory Architectural Variants and Additional Experimental Results

To better understand the design space of text-to-audio generation for respiratory sounds, we conducted exploratory experiments evaluating alternative architectural modifications beyond the proposed memory and watermarking framework. These include squeeze-and-excitation (SE) encoders and channel-wise gating mechanisms. Table B.1 summarizes the performance of these variants. The results show that these modifications do not provide consistent improvements in FAD and KL, indicating limited gains in aligning generated audio with real respiratory data. For example, adding SE encoding increases FAD from 6.73 to 7.96, while combining SE with channel-wise gating results in further shifts in performance (FAD = 9.19, KL = 6.18). We also evaluate removing these components from the proposed model. The results indicate that these configurations do not lead to improvements over the full model. Overall, these findings suggest that architectural techniques commonly used in general audio modeling may not directly translate to respiratory sound generation, while memory-based conditioning provides a more stable and effective solution.

Table B.1: Exploratory experiments evaluating alternative architectural modifications. These configurations do not consistently improve performance compared to the proposed memory and watermarking design.

| Configuration | FAD ↓ | KL ↓ | IS ↑ | PRISM ↑ |
|---|---|---|---|---|
| **Baseline (No Memory, No Watermarking)** | 6.73 | 0.67 | **1.39** | 0.21 |
| **Memory + Watermarking (Proposed)** | **2.72** | **0.50** | 1.22 | **0.23** |
| *Additional Modifications* | | | | |
| + SE Encoder | 7.96 | 5.92 | 1.03 | 0.14 |
| + Channel-Wise Gating | 8.86 | 7.42 | 0.55 | 0.13 |
| + SE Encoder + Channel-Wise Gating | 9.19 | 6.18 | 0.58 | 0.13 |
| *Component Variants* | | | | |
| − SE Encoder | 9.71 | 8.33 | 0.84 | 0.13 |
| − Channel-Wise Gating | 9.73 | 6.53 | 0.57 | 0.20 |

# C PRISM Validation (20 recordings)

**Section moved to appendix**

We test PRISM using 20 recordings from the ground-truth dataset under three pairing settings: identical pairs (GT vs same GT), perturbed pairs (GT vs slightly modified GT), and mismatched pairs (GT vs different GT). As expected, identical pairs produce the highest scores, perturbed pairs fall in between, and mismatched pairs give the lowest values. This trend is shown in Figure C.1. The behavior indicates that PRISM responds to both signal distortion and structural differences in the audio, rather than only overall similarity.

# D PRISM Weights Study

**Figure moved to appendix**

# E Downstream Classification Setup and Design

We evaluate the generated data using two downstream pipelines: a VGGish-based model operating on log-Mel spectrogram patches with attention pooling, and an Audio Spectrogram Transformer (AST) model that captures longer-range temporal dependencies. Both are trained and evaluated under consistent settings to assess the impact of synthetic data across different modeling approaches.

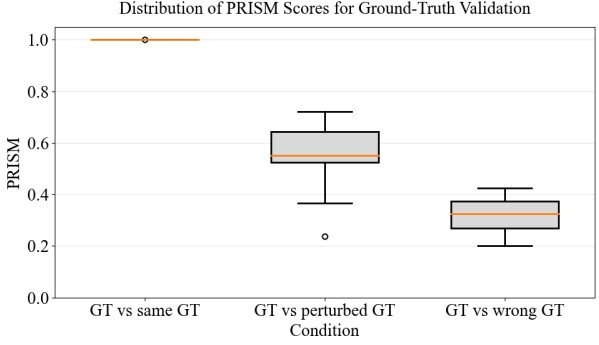
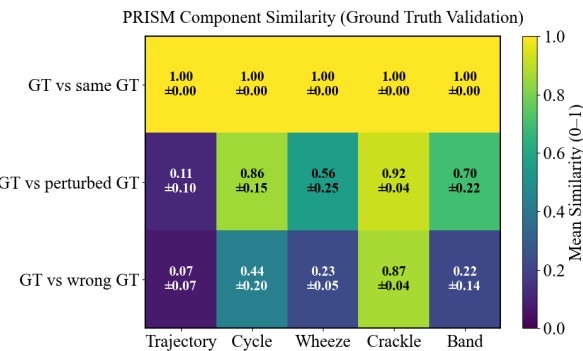

(a) Distribution of overall PRISM scores across controlled pairing conditions.

(b) Component-wise PRISM similarity (mean ± standard deviation across sample pairs).

Figure C.1: PRISM validation on 20 ground-truth recordings under controlled conditions. Left, identical pairs achieve the highest scores, with decreasing values under perturbation and mismatch. Right, component-wise similarity shows consistent trends across trajectory, cycle, wheeze, crackle, and band features.

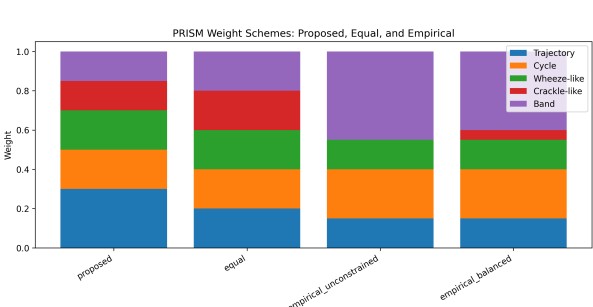
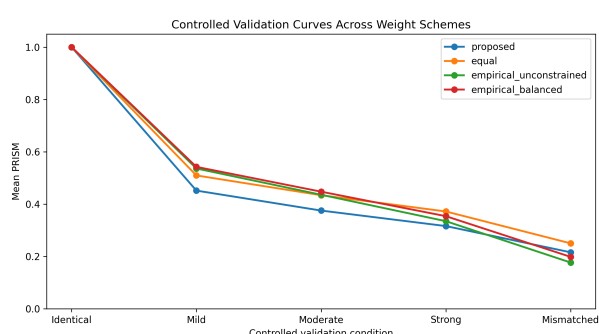

(a) Component weights for the proposed heuristic, equal, and empirically selected PRISM schemes.

(b) Validation behaviour under each weighting scheme.

Figure D.1: PRISM weight-sensitivity study. Left: component weights under the proposed, equal, and empirical schemes. Right: all schemes preserve the expected ordering, with identical pairs scoring highest, followed by mild, moderate, and strong perturbations, and mismatched pairs scoring lowest.

### E.1 VGGish-Based Downstream Model

To support the downstream results reported in Table 5, we provide the detailed training and evaluation setup of the VGGish-based classifier (Han et al., 2022). The model operates on log-Mel spectrogram patches extracted from cough recordings and aggregates patch-level features using attention-based pooling. Each waveform $x$ is resampled to 16 kHz, converted to mono, and normalized to a fixed duration. A log-Mel spectrogram is computed and scaled to the range $[-1, 1]$:

$$\mathbf{S} = \text{MelSpec}(x), \quad \tilde{\mathbf{S}} = 2 \cdot \frac{\mathbf{S} - \min(\mathbf{S})}{\max(\mathbf{S}) - \min(\mathbf{S})} - 1. \tag{E.1}$$

This transformation ensures a consistent dynamic range across all samples.

The spectrogram is divided into fixed-width patches:

$$\tilde{\mathbf{S}} \rightarrow \{\mathbf{P}_1, \dots, \mathbf{P}_T\}, \tag{E.2}$$

where each patch captures a local time–frequency segment of the cough signal.

Each patch is encoded using the VGGish backbone:

$$\mathbf{h}_t = f_{\text{VGGish}}(\mathbf{P}_t), \quad \mathbf{h}_t \in \mathbb{R}^{512}. \tag{E.3}$$

Table E.1: VGGish-based downstream classification setup.

| Component | Setting |
|---|---|
| *Model Architecture* | |
| Backbone | VGGish-style CNN |
| Feature dimension | 512 |
| Pooling | Attention-based pooling |
| Classifier | MLP ($512 \rightarrow 128 \rightarrow 64 \rightarrow 1$) |
| Dropout | 0.5 |
| *Input Processing* | |
| Sampling rate | 16 kHz |
| Audio duration | 10 seconds |
| Channels | Mono |
| FFT size | 1024 |
| Window length | 400 samples |
| Hop length | 160 samples |
| Mel bins | 64 |
| Representation | Log-Mel spectrogram (scaled to [-1,1]) |
| Patch width | 96 frames |
| Patch extraction | Non-overlapping (stride = 96) |
| *Training Setup* | |
| Batch size | 32 |
| Epochs | 50 |
| Optimizer | AdamW |
| Learning rate (backbone) | $1 \times 10^{-5}$ |
| Learning rate (head) | $1 \times 10^{-3}$ |
| Weight decay | $1 \times 10^{-3}$ |
| Loss | Binary focal loss ($\alpha = 0.85$, $\gamma = 2.0$) |
| Class balancing | Weighted random sampling (inverse frequency) |
| Scheduler | CosineAnnealingWarmRestarts |
| *Augmentation* | |
| Audio augmentations | Noise, time stretch, pitch shift, temporal shift |
| GAN augmentation | LSGAN patch replacement (p = 0.6, positives only) |
| GAN epochs | 35 |
| Latent dimension | 128 |
| *Evaluation* | |
| Threshold selection | Validation-based (Youden's J) |
| Metrics | F1, AUC, Sensitivity, Specificity |

These embeddings represent local acoustic features such as spectral shape and temporal energy.

Patch-level features are aggregated using attention. The attention score for each patch is computed as

$$a_t = \mathbf{w}_2^\top \tanh(\mathbf{W}_1 \mathbf{h}_t + \mathbf{b}_1), \tag{E.4}$$

and converted into normalized weights

$$\alpha_t = \frac{\exp(a_t)}{\sum_j \exp(a_j)}, \tag{E.5}$$

which indicate the relative importance of each patch.

The final representation is obtained by weighted pooling:

$$\mathbf{c} = \sum_t \alpha_t \mathbf{h}_t. \tag{E.6}$$

This produces a single vector summarizing the entire cough recording.

The classifier predicts the probability of the positive class:

$$\hat{y} = \sigma(f_{\mathrm{cls}}(\mathbf{c})), \tag{E.7}$$

where $\sigma$ is the sigmoid function.

Training uses binary focal loss:

$$\mathcal{L} = -\alpha y(1 - \hat{y})^\gamma \log(\hat{y}) - (1 - \alpha)(1 - y)\hat{y}^\gamma \log(1 - \hat{y}), \tag{E.8}$$

which emphasizes hard examples and mitigates class imbalance.

Synthetic augmentation is introduced using a generator:

$$\tilde{\mathbf{P}} = G(\mathbf{z}), \quad \mathbf{z} \sim \mathcal{N}(0, I), \tag{E.9}$$

where generated patches replace real positive patches during training to increase diversity.

The generator and discriminator are trained using least-squares objectives:

$$\mathcal{L}_D = \frac{1}{2}(D(\mathbf{P}) - 1)^2 + \frac{1}{2}D(G(\mathbf{z}))^2, \tag{E.10}$$

$$\mathcal{L}_G = (D(G(\mathbf{z})) - 1)^2. \tag{E.11}$$

These losses stabilize training and improve the quality of generated patches.

During evaluation, the decision threshold is selected from the ROC curve:

$$J(\tau) = \mathrm{TPR}(\tau) - \mathrm{FPR}(\tau), \tag{E.12}$$

$$\tau^* = \arg\max_\tau J(\tau), \tag{E.13}$$

which maximizes the balance between sensitivity and specificity.

### E.2 Asthma Classification Experimental Setup

**New section for the Asthma downstream experiment** This section describes the classifier setup, the training data construction, and the augmentation settings used for the asthma classification experiment.

#### E.2.1 Training and Preprocessing Setup

The same preprocessing and model configuration were used for all methods. Each input recording was converted to mono, resampled to 16 kHz, and cropped or padded to 10 seconds. The audio was then converted into a log-mel spectrogram and passed to a VGGish-based classifier.

#### E.2.2 Training Data

The training set was balanced for all methods. Each experiment used 5,884 negative samples and 5,884 positive samples, giving a total of 11,768 training samples. The validation and test sets were unchanged and contained only real recordings.

#### E.2.3 Classical Augmentation Settings

Classical augmentation was applied to asthma-positive recordings before training. The augmented recordings were then used as additional positive samples in the training set.

Table E.2: Training and preprocessing settings.

| Setting | Value |
|---|---|
| Task | Binary asthma classification |
| Input | Three-cough recording |
| Sampling rate | 16 kHz |
| Audio length | 10 seconds |
| Channels | Mono |
| Input representation | Log-mel spectrogram |
| Mel bins | 64 |
| FFT size | 1024 |
| Window length | 400 samples |
| Hop length | 320 samples |
| Spectrogram size | $64 \times 500$ |
| Normalisation | Per-sample mean/std normalisation |
| Model | VGGish-based classifier |
| Classifier head | 512–256–64–1 |
| Dropout | 0.6 |
| Loss | Binary focal loss |
| Focal loss $\alpha$ | 0.75 |
| Focal loss $\gamma$ | 2.0 |
| Optimiser | AdamW |
| Backbone learning rate | $1 \times 10^{-5}$ |
| Classifier learning rate | $1 \times 10^{-3}$ |
| Weight decay | $1 \times 10^{-2}$ |
| Batch size | 32 |
| Maximum epochs | 50 |
| Early stopping patience | 8 epochs |
| Model selection | Best validation AUC |
| Threshold selection | Validation G-mean |
| Threshold range | 0.10–0.89, step 0.01 |
| Random seed | 42 |

Table E.3: Training data used for each method.

| Method | Real Negatives | Real Positives | Generated/Augmented Positives | Total |
|---|---|---|---|---|
| No Augmentation | 5,884 | 5,884 repeated | 0 | 11,768 |
| GAN Augmentation | 5,884 | 5,884 repeated | GAN patch injection | 11,768 |
| Classical Augmentation | 5,884 | 2,942 | 2,942 | 11,768 |
| LungTTA Augmentation | 5,884 | 2,942 | 2,942 | 11,768 |

### E.3 Audio Spectrogram Transformer (AST) Model

To evaluate the practical utility of the generated audio, we design a downstream classification task using an Audio Spectrogram Transformer (AST) backbone (Gong et al., 2021). The model extracts mid-layer representations (block 7), which provide a balance between local acoustic features and higher-level temporal semantics. The full setup is summarized in Table E.5.

Each waveform $x$ is resampled to 16 kHz, converted to mono, high-pass filtered, and padded or truncated to a fixed duration before being processed by the AST feature extractor.

Table E.4: Classical waveform augmentation settings.

| Operation | Range | Probability |
|---|---|---|
| Gaussian noise | Amplitude 0.001–0.015 | 0.5 |
| Time stretch | Rate 0.8–1.25 | 0.5 |
| Pitch shift | $-4$ to $+4$ semitones | 0.5 |
| Time shift | $-0.5$ to $+0.5$ | 0.5 |

Table E.5: AST-based downstream classification setup.

| Component | Setting |
|---|---|
| *Model Architecture* | |
| Backbone | AST (Audio Spectrogram Transformer) |
| Feature layer | Block 7 (CLS token) |
| Feature dimension | 768 |
| Classifier | Linear $(768 \rightarrow 1)$ |
| Dropout | 0.65 |
| *Input Processing* | |
| Sampling rate | 16 kHz |
| Audio duration | 6 seconds |
| Channels | Mono |
| High-pass filter | 50 Hz |
| Silence trimming | Energy-based (top_db $= 30$) |
| Representation | AST feature extractor (log-Mel based) |
| *Training Setup* | |
| Batch size | 32 |
| Optimizer | AdamW (SAM for training phase) |
| Learning rate (backbone) | $2 \times 10^{-6}$ |
| Learning rate (head) | $2 \times 10^{-5}$ |
| Weight decay | 0 |
| Scheduler | Cosine annealing |
| Loss | Binary cross-entropy (logits) |
| *Augmentation* | |
| Feature augmentation | Time masking, frequency masking |
| Mix up | Enabled (probabilistic) |
| *Evaluation* | |
| Test-time augmentation | 3 views (original, $\pm500$ ms shift) |
| Prediction aggregation | Mean over views |
| Threshold selection | Validation-based (Youden's J) |
| Metrics | F1, AUC, Sensitivity, Specificity |

We adopt a mid-layer feature extraction strategy. Let $\mathbf{H}^{(l)}$ denote the hidden representation at transformer block $l$. The feature vector is extracted from block 7 using the [CLS] token:

$$\mathbf{h} = \mathbf{H}_{\text{CLS}}^{(7)} = \text{AST}_{\text{block 7}}(x)_{\text{CLS}}. \tag{E.14}$$

This representation captures both local spectral structure and longer-range temporal dependencies.

The feature is regularized using dropout:

$$\tilde{\mathbf{h}} = \text{Dropout}(\mathbf{h}, p = 0.65), \tag{E.15}$$

The classifier computes the logit:

$$z = \mathbf{W}\tilde{\mathbf{h}} + b, \tag{E.16}$$

and predicts the probability of the positive class:

$$\hat{y} = \sigma(z), \tag{E.17}$$

Training is performed using binary cross-entropy with logits:

$$\mathcal{L} = -\left[y \log \sigma(z) + (1 - y) \log \left(1 - \sigma(z)\right)\right]. \tag{E.18}$$

To improve robustness, test-time augmentation is applied by generating multiple shifted views of the same input:

$$\hat{y}^{(i)} = f(x^{(i)}), \tag{E.19}$$

and aggregating predictions:

$$\hat{y}_{\text{final}} = \frac{1}{N} \sum_{i=1}^{N} \hat{y}^{(i)}. \tag{E.20}$$

The decision threshold is selected on the validation set:

$$J(\tau) = \text{TPR}(\tau) - \text{FPR}(\tau), \tag{E.21}$$

$$\tau^* = \arg\max_{\tau} J(\tau), \tag{E.22}$$

and applied unchanged during test evaluation.

## F  Watermark study

**New figure**

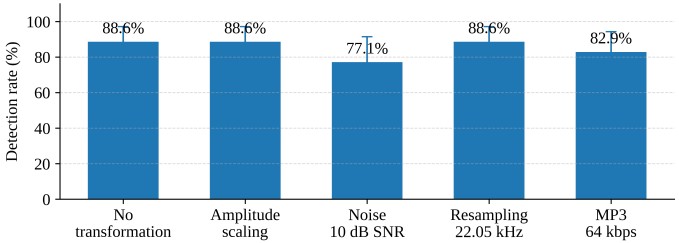

Figure F.1: Watermark detection under representative audio transformations.

## G  Text Prompts

Prompts are constructed from metadata fields available in the respiratory datasets, including age, sex, smoking status, and respiratory conditions such as asthma and COPD. These attributes are converted into simple text descriptions that specify the target sound and its context. The prompts provide the main conditioning signal during generation, while metadata embeddings and the memory module are used as additional inputs within the model.

### G.1 Main Evaluation Prompts

A sustained /e/ vowel sound from a 57-year-old female smoker with possible asthma and no COPD, with reduced breath support.

A dry cough from a 65-year-old male smoker with no asthma and no COPD.

A cough from a 50-year-old female non-smoker with possible asthma and no COPD.

A cough from a 74-year-old female smoker with no asthma and no COPD.

A cough from an 18-year-old female non-smoker with possible asthma and no COPD.

A sustained /a/ vowel sound from a 20-year-old male non-smoker with no asthma and no COPD.

A sustained /o/ vowel sound from a 60-year-old male smoker with no asthma and no COPD.

A cough from a 77-year-old male smoker with no asthma and no COPD.

A cough from a 60-year-old male smoker with possible asthma and possible COPD, with slightly impaired airflow.

A cough from a 73-year-old male smoker with no asthma and no COPD, with reduced breath strength.

A cough from a 58-year-old male smoker with asthma and COPD, with heavy chest involvement.

A cough from a 48-year-old male smoker with no asthma and no COPD.

A cough from a 65-year-old male smoker with COPD, with obstructed airflow characteristics.

A sustained /a/ vowel sound from a 24-year-old male non-smoker with no asthma and no COPD.

A cough from a 30-year-old female smoker with asthma and no COPD.

A cough from an 80-year-old male smoker with COPD, with weak respiratory effort.

Shallow breathing from a 54-year-old female smoker with no asthma and no COPD.

Deep breathing from a 37-year-old female smoker with no asthma and no COPD.

A cough from an 18-year-old female non-smoker with no asthma and no COPD.

A cough from a 65-year-old male smoker with COPD, with reduced airflow.

### G.2 Example Prompt Variations

A dry cough from a female smoker aged 30–40 years. A wet cough from a male non-smoker aged 40–50 years. Shallow coughing from a person with asthma. A heavy cough from an elderly person.

## H Survey Design

Participants listened to each audio sample using an embedded audio player and rated it using a questionnaire. Each sample was evaluated independently on a 5-point Likert scale. The interface displayed the waveform alongside the audio. General listeners rated overall quality and alignment with the prompt, while clinical experts additionally rated clinical usefulness. The scales were designed to separate perceptual quality from clinical relevance.

### H.1  General Listener Evaluation (Example)

**Audio Sample:**
"A sustained /a/ vowel sound produced by a 24-year-old male non-smoker with no asthma and no COPD"

**Overall Quality (OVL)**
(Consider clarity, naturalness, and overall sound quality.)

| Very Poor | Poor | Fair | Good | Excellent |
|---|---|---|---|---|

**Relevance to Text Input (REL)**
How well does this audio match the description?

| Not Relevant at All | Slightly Relevant | Moderately Relevant | Very Relevant | Highly Relevant |
|---|---|---|---|---|

### H.2  Expert Clinical Evaluation (Example)

**Audio Sample:**
"A cough from a 60-year-old male smoker with asthma and COPD, with impaired airflow"

**Overall Quality (OVL)**
(Consider clarity, naturalness, and overall sound quality.)

| Very Poor | Poor | Fair | Good | Excellent |
|---|---|---|---|---|

**Relevance to Text Input (REL)**
How well does this audio match the description?

| Not Relevant at All | Slightly Relevant | Moderately Relevant | Very Relevant | Highly Relevant |
|---|---|---|---|---|

**Clinical Relevance for Assessment (CRA)**
(Consider whether the audio contains meaningful respiratory patterns such as realistic cough structure, airflow limitation, and clinically interpretable acoustic features.)

| Not Clinically Useful at All | Slightly Useful | Moderately Useful | Very Useful | Highly Clinically Useful |
|---|---|---|---|---|

