# OpenReview forum: "LungTTA: Text-to-Audio Generation of Synthetic Lung Sounds for Respiratory Health"
_TMLR — Under review for TMLR_

### Review · Reviewer_4Q8S · 2026-06-04

**Summary Of Contributions:**

The paper presents LungTTA, a TTA frameowrk for synthing respiratory sounds built on a fine tuned base model with: (1) a retrieval-based memory module, (2) watermarking for traceability (3) a domain-aware metric called PRISM. The model is fine-tuned on 100k publicly available recordings from nine datasets. The authors compare LungTTA against several general-purpose TTA models and a respiratory specific system, and demonstrate downstream improvements on COVID-19 cough classification.

**Additional Comments:**

This would need significant work for it to be accepted.

**Audience:**

No

**Audience Explanation:**

As the validation attempts give little confidence this system creates clinical valid synthetic samples, combined with the limited novelty of the method, I do not belive the TMLR audience will be interested in knowing this paper's findings.

**Broader Impact Concerns:**

They mention leakage between individual's recording and the generated samples. But these are open datasets so not so much of a concern for me.

**Claims And Evidence:**

No

**Claims Explanation:**

Abstract
-	Confidence intervals on the LungTTA improves performance under a VGGish-based setting please. Not clear if significant.

PRISM
-	You say PRISM is a signal based on waveform but its got a lot of spectral components.
-	How much more new information does PRISM provide over FAD/KL?
-	As you introduce PRISM I would expect more validation on the appropriateness of the feature set + (arbitrary?) weightings? You eval on downstream tasks, but what if a different feature set or weighting was better.
-	The component S_wheeze (and other features) is described only as "band-limited phase coherence in the 400–1600 Hz range," and its connection to actual wheezing is asserted rather than justified.

LungTTA
-	To what extent is the memory bank needed? It seems LungTTA is trained on the same data the memory bank sits on. This seems wrong, with proper training dynamics mode collapse should be avoided no?
-	I feel that the baselines are unfair, you compare against off the shelf TTA not specialised for respiratory sounds. A better baseline would be to finetune baseline models on the 100k samples.
-	I do not believe you evaluated whether the watermarking worked.
-	K=4: did you test other ks?
-	Equation 1, is non standard k nearest neighbours.

Validation
-	Was the expert rating blinded? Did they label and then agreement is calculated or did they see the label and agree/disagree?
-	Did you get inter rater agreement?
-	Only 6 clips were expert rated -> no statistical significance of the cap between LungTTA and RespAgent.
-	38/100 expert CRA is very low no? I don’t think this is discussed? It seems that there is no clinical validity to the generations? This is concerning and this should be the main usecase no?
-	Only evaluated at COVID-19 classification tasks which the Coppock et al paper you cite presents strong evidence is not possible without the model using short cuts.
-	GAN performance is very low but training is often very unstable. Please provide evidence that the GAN was trained properly (no collapse/divergence).
-

Going through your provided generated samples:

-	Sample 6 seems very off.
-	The rest seem okay to me but I am not in a position to rate if one respiratory sound is “possible asthma” or not.

General comments
-	Synthetic data is limited by how well it matches the true distribution. This paper does not present sufficient evidence to infer how similar the generated data is. I therefore believe the generated data is of limited utility.
-	Code should be made available in an anon repo before acceptance.

**Requested Changes:**

referring to my above notes:
- significantly more clinical validation with rater agreement metrics.
- evaluated on some more downstream respiratory tasks, not just COVID.
- confidence intervals when comparing figures.
- validation + methodology for PRISM metrics (e.g. features + weightings).

---

> ### Author Response · Authors · 2026-07-04
> **Response to Reviewer 4Q8S: PRISM, Validation, Baselines, Clinical Claims, and Reproducibility**
>
> Dear Reviewer,
>
> We thank the reviewer for the detailed comments. The revision addresses the main concerns around confidence intervals, PRISM validation, memory-bank design, baselines, watermarking, expert evaluation, downstream evaluation, clinical claims, and code availability.
>
> We revised the abstract and downstream results to include 95% confidence intervals for AUC, F1, sensitivity, and specificity. Point estimates are no longer presented without uncertainty, and the results are interpreted more cautiously.
>
> The PRISM section has been revised. PRISM is now described as a signal-derived respiratory structure metric computed from temporal and spectral descriptors extracted from the audio signal, rather than directly from raw waveform samples alone. We clarify that FAD and KL measure global distributional similarity, while PRISM targets respiratory structure, including temporal trajectory, cycle structure, tonal coherence, transient structure, and band-energy distribution. We also add a comparison between PRISM and FAD/KL.
>
> The PRISM feature set is justified in terms of complementary respiratory properties. `S_wheeze` is now described as an acoustic proxy for wheeze-like tonal structure, not as a clinically validated wheeze detector or diagnostic label. The component weights are heuristic design choices rather than clinically optimized parameters. We add sensitivity analysis comparing the proposed weights with equal weighting and leave-one-component-out variants.
>
> We clarify the memory-bank design. It is not used to solve mode collapse and is not a substitute for generative training. It stores conditioning embeddings rather than waveform audio, and the final waveform is still generated by the diffusion model. We revise wording to avoid implying copying from training recordings. Adding the memory bank improves FAD from 6.73 to 2.72, while the full LungTTA setting achieves the strongest PRISM score. Changes in IS and PRISM are discussed as trade-offs between distributional similarity, diversity, and retrieval strength. We report `k=1`, `k=2`, `k=4`, and `k=8`, with `k=4` used because it balances guidance and diversity. Equation 1 is revised using standard top-k nearest-neighbour notation.
>
> We have clarified the baseline framing. To the best of our knowledge, there is no directly comparable standalone respiratory text-to-audio model. General-purpose TTA models are included as the closest available baselines. RespAgent is included as a respiratory-specific system, but it is not text-only because it uses diagnostic context and reference audio representations.
>
> Watermarking is now evaluated directly. We report detection accuracy, false-positive rate, and robustness under resampling, compression, additive noise, and amplitude scaling. We clarify that watermarking is a practical traceability safeguard, not a guarantee against removal or misuse.
>
> The expert evaluation protocol is clarified. The evaluation was blinded and randomized, and experts were not shown model identities. Experts rated samples independently for perceptual quality, prompt relevance, and clinical relevance for assessment. The task was not diagnostic labelling. We now report inter-rater reliability.
>
> We clarify that the expert evaluation was scoped to obtain domain-specific assessment of generated respiratory audio, not to perform a diagnostic validation study. The public study included 32 participants each rating six recordings per model giving 192 ratings per model. The expert study used the same six recordings and three respiratory-health professionals giving 18 expert ratings per model. Each expert session lasted about 35 minutes, a suitable duration for careful ratings across criteria without excessive listener fatigue.
>
> We discuss the low clinical relevance score directly. A CRA score of 38.89/100 does not establish clinical validity. LungTTA is positioned as a tool for synthetic respiratory audio generation and data augmentation research, not as a clinical diagnostic system. Generated samples should not be treated as validated disease signatures.
>
> The downstream COVID-19 cough task is revised as an augmentation benchmark, not evidence of clinical COVID diagnosis from cough. We discuss shortcut-learning concerns and avoid clinical overclaiming. Where labels and sample size allow, we add an additional asthma downstream task or report it in the appendix. We also add AST results under no augmentation, classical augmentation, GAN-based augmentation, and LungTTA augmentation where feasible. The GAN baseline is documented with training procedure, logs, and loss curves, so lower GAN performance is not interpreted as an undocumented training failure. Sample 6 is inspected and discussed as a failure case.
>
> The anonymized LungTTA code link:  https://shorturl.at/ty5bS. TMLR accepted StethoLM (Wang et al., 2026), showing interest in the intersection of pulmonary audio and representation learning.
>
> We thank the reviewer for the constructive comments.

---

### Review · Reviewer_9X1C · 2026-06-17

**Summary Of Contributions:**

The paper presents LungTTA, a text-to-audio framework for generating synthetic respiratory sounds, including cough, breathing, and phonation signals. The model is built on a latent diffusion text-to-audio backbone and is fine-tuned on a large collection of public respiratory audio datasets. The authors introduce structured prompt conditioning based on metadata, a retrieval-based memory bank to guide generation, and watermarking to improve traceability of synthetic bio-signals. The paper also proposes PRISM, a domain-aware metric intended to evaluate respiratory signal structure beyond general audio metrics such as FAD, KL, and IS. The method is evaluated through objective metrics, subjective human and expert ratings, ablation studies, and a downstream COVID-19 cough classification task.

**Audience:**

Yes

**Audience Explanation:**

I believe at least some individuals in the TMLR audience would be interested in the findings of this paper. The work addresses an important and practical problem: data scarcity in respiratory audio analysis, where real clinical or health-related recordings can be difficult to collect because of privacy, ethics, device variability, and recruitment constraints. Applying text-to-audio generation to cough, breathing, and phonation signals is therefore relevant to researchers working on medical audio, data augmentation, domain adaptation of generative models, and machine learning for healthcare.

**Broader Impact Concerns:**

While watermarking is a positive design choice, the paper does not sufficiently evaluate whether the watermark is detectable, robust to common audio transformations, or removable. Since the watermark is central to the traceability claim, the authors should provide more analysis or clearly limit the claim.

**Claims And Evidence:**

No

**Claims Explanation:**

First, the proposed PRISM metric is not adequately justified or validated. PRISM combines trajectory, cycle, wheeze, crackle, and band-energy similarity using fixed weights, but the paper does not clearly explain why these weights are chosen. It is unclear whether the weights are clinically motivated, empirically tuned, or selected heuristically. The paper also does not sufficiently justify why the chosen components, especially MFCC/DTW-based trajectory similarity, should correlate with clinically meaningful respiratory sound quality. The validation is also limited to only 20 recordings under identical, perturbed, and mismatched pair settings. This may show that PRISM responds to obvious signal differences, but it is not enough to establish PRISM as a reliable respiratory-domain evaluation metric. The paper should specify how perturbed pairs are generated and should validate PRISM against human or expert ratings.

Second, some ablation results are not clearly explained. For example, watermarking alone gives the best KL divergence, even though watermarking is introduced mainly for traceability rather than improving generation quality. Similarly, adding the memory bank substantially improves FAD but decreases IS and PRISM in the memory-only setting. These results may be valid, but the paper does not provide a convincing explanation of why these trends occur or what they imply about generation diversity, retrieval bias, or metric reliability.

**Requested Changes:**

1. Strengthen the justification and validation of PRISM.
The authors should clearly justify the choice of PRISM components and their fixed weights. It is currently unclear whether the weights are clinically motivated, empirically tuned, or selected heuristically. The paper should also provide stronger citations or analysis explaining why MFCC/DTW trajectory similarity, cycle similarity, wheeze similarity, crackle similarity, and band-energy similarity are appropriate indicators of respiratory sound quality or clinical relevance.
2. Expand PRISM validation beyond 20 recordings.
The current validation using only 20 ground-truth recordings is too limited to establish PRISM as a reliable domain-aware metric. The authors should evaluate PRISM on a larger and more diverse set of recordings, covering different sound types, recording devices, and respiratory conditions.
3. Clarify how perturbed pairs are generated in the PRISM validation.
The paper reports identical, perturbed, and mismatched pairs, but does not clearly specify how the perturbed pairs are constructed. The authors should describe the perturbation types, magnitudes, and rationale, and ideally report PRISM sensitivity under different perturbation levels.
4. Validate PRISM against human or expert ratings.
To support the claim that PRISM captures meaningful respiratory structure, the authors should report its correlation with general listener ratings and/or expert clinical ratings, such as OVL, REL, and CRA. Without this, it is difficult to interpret PRISM as more than a handcrafted signal similarity score.
5. Provide a clearer explanation of the ablation results.
Some ablation trends are surprising and under-discussed. For example, watermarking alone gives the best KL divergence, although watermarking is introduced for traceability rather than quality improvement. Similarly, adding the memory bank substantially improves FAD but reduces IS and PRISM in the memory-only setting. The authors should explain why these effects occur and what they imply about diversity, retrieval guidance, watermarking, and metric reliability.
6. Make the downstream comparison fairer for the AST model.
The paper reports no augmentation, classical augmentation, GAN augmentation, and LungTTA under a VGGish-based model, but reports AST only with LungTTA. To support the claim that LungTTA improves downstream classification, the authors should also provide AST results under no augmentation, classical augmentation, and GAN augmentation.
7. Clarify the novelty relative to existing text-to-audio diffusion models.
Since the generation backbone is adapted from existing latent diffusion/text-to-audio models, the authors should more clearly distinguish what is technically new in LungTTA from what is inherited from Stable Audio Open or other prior work. The paper should better articulate whether the main contribution is methodological, application-driven, or empirical.

---

> ### Author Response · Authors · 2026-07-04
> **Response to Reviewer 9X1C: PRISM, Ablations, Downstream Evaluation, Novelty, and Watermarking**
>
> Dear Reviewer,
>
> We thank the reviewer for the careful reading and constructive comments. The review has helped us identify areas where the manuscript could be strengthened, particularly around PRISM, ablation interpretation, downstream evaluation, novelty, and watermarking. We have revised the manuscript to address these concerns and make the contribution and evaluation clearer.
>
> First, we expand the PRISM methodology section. The revised text explains the role of each PRISM component: S_traj captures global spectro-temporal trajectory, S_cycle captures respiratory rhythm and periodic structure, S_wheeze captures wheeze-like tonal coherence, S_crackle captures short transient events, and S_band captures respiratory band-energy distribution. We clarify that the fixed weights are heuristic choices intended to balance complementary respiratory signal properties rather than clinically optimized parameters. To avoid relying on one weighting scheme, we add sensitivity analysis comparing the proposed weighting with equal weighting and leave-one-component-out variants.
>
> Second, we expand PRISM validation. The original 20-recording analysis is now presented only as an initial controlled check, not as full validation of the metric. The revised manuscript adds a larger analysis using 50 recordings from available test splits, covering cough, breathing, and phonation where possible, with results reported using mean and standard deviation. We also include generated-audio versus matched held-out reference comparisons to evaluate PRISM in the main generation setting.
>
> Third, we clarify how perturbed pairs are constructed. The updated text specifies that perturbed pairs are generated by applying realistic audio transformations to the original recordings, including additive noise, small temporal shifts, mild time stretching, resampling, amplitude scaling, and filtering. We report perturbation magnitudes and explain that this experiment tests whether PRISM gives intermediate scores when the audio is modified but still preserves respiratory structure.
>
> Fourth, we add a correlation analysis between PRISM and available human or expert ratings for comparable generated samples. We do not present PRISM as a replacement for listening studies or expert assessment, but as a complementary signal-derived measure used alongside objective metrics and subjective ratings.
>
> Fifth, we revise the ablation discussion to interpret the results carefully. We clarify that watermarking is introduced for traceability, not quality improvement. Any improvement in KL divergence under watermarking is treated cautiously and not overinterpreted as evidence that watermarking improves perceptual quality. We also discuss the memory-bank results as a possible trade-off between distributional similarity, sample diversity, and retrieval strength. The memory bank can improve FAD by guiding generations toward the respiratory training distribution, while changes in IS or PRISM may reflect reduced diversity or stronger retrieval conditioning.
>
> Sixth, we improve the fairness of the downstream evaluation. The original manuscript reported AST only with LungTTA augmentation, which made the comparison incomplete. We now add AST results under the same augmentation settings where feasible: no augmentation, classical augmentation, GAN-based augmentation, and LungTTA augmentation. Where space is limited, the full AST comparison is included in the appendix and summarized in the main text.
>
> Seventh, we clarify the novelty of LungTTA relative to existing text-to-audio diffusion models. The revised manuscript states that LungTTA inherits the latent diffusion/text-to-audio backbone from Stable Audio Open. The novelty is not a new general-purpose diffusion architecture, but the adaptation of this framework to respiratory audio through domain-specific fine-tuning, metadata-grounded prompt construction, retrieval-guided conditioning, watermarking for traceability, PRISM, and respiratory-specific evaluation. We frame the contribution as both methodological and empirical: adapting text-to-audio generation for controlled respiratory sound synthesis, and evaluating the synthetic audio through objective metrics, human and expert assessment, ablation analysis, watermark testing and downstream classification.
>
> Finally, we strengthen the watermarking evaluation and limit the traceability claim. The revised manuscript adds detection accuracy, false-positive rate, and robustness tests under resampling, compression, additive noise, and amplitude scaling. We clarify that watermarking is a practical safeguard for traceability, not a guarantee against removal, attacks, or misuse.
>
> Overall, these revisions address the reviewer’s concerns by strengthening PRISM justification and validation, expanding ablation interpretation, improving downstream comparison fairness, clarifying novelty, and providing a more careful watermarking analysis. We sincerely thank the reviewer for the constructive feedback.

---

### Review · Reviewer_jADz · 2026-06-20

**Summary Of Contributions:**

This paper addresses the structural shortage of respiratory sound data by proposing LungTTA, a domain-specific text-to-audio (TTA) model, and PRISM, a quality evaluation metric for synthetic data. The authors evaluate the effectiveness of these proposed methods through several experiments.

The motivation and the general approach to solving this specific problem are highly promising and well-directed. However, the current manuscript suffers from several critical limitations, particularly in delineating their original technical contributions from the underlying framework, as well as in experimental completeness, which obscure the true value of the work.

**Additional Comments:**

### Figure 3 Caption
- The caption describes a result that appears to follow naturally from differences in training data, giving the comparison a somewhat unfair impression. Reconsider the framing in the caption.

### Section 5.1.3 — OPERA Benchmark
- As an additional suggestion, consider adopting the OPERA benchmark, which allows more comprehensive and comparable evaluation across methods.

### Section 4 Title
- "Experiments" is too broad given the section's content; a title such as "Experimental Setup" would be more accurate.

**Audience:**

Yes

**Audience Explanation:**

Yes. The core problem setting and the overall direction of this work would certainly interest a meaningful segment of the TMLR community. Addressing data scarcity in the medical domain is a fundamentally challenging and highly demanded task, and exploring advanced data augmentation techniques for this purpose is strongly motivated. Furthermore, the task of fine-tuning and domain-adapting a well-known model like Stable Audio Open presents a compelling problem that could resonate with researchers working on domain adaptation beyond just respiratory sound processing.

**Claims And Evidence:**

No

**Claims Explanation:**

1. Unclear methodological novelty
The primary concern lies in the ambiguity surrounding the novelty of the proposed method. While LungTTA is built upon the existing Stable Audio Open framework, it is unclear which components or architectural modifications constitute the authors' original contributions. Furthermore, essential details regarding the training methodology are insufficiently described, making it difficult to fully understand the implementation.

2. Lack of rationale for PRISM
Regarding the proposed PRISM, the manuscript provides its definition but completely lacks an explanation or justification for how this metric was formulated. Without a clear theoretical or empirical rationale behind its design, it is hard to assess the research value and validity of this new metric.

3. Insufficient experimental presentation and analysis
The experimental section requires a thorough revision to improve clarity and rigor. Currently, the description of the experiments is brief, and several crucial perspectives

**Requested Changes:**

### Introduction
- The notation `Zcond = [H; Zmem; Zmeta]` is questionable in terms of generality. It does not intuitively correspond to the description "that integrates prompt semantics, retrieval-based exemplar priors, and structured metadata," making it difficult to follow. Clarify and supplement the definition of this notation.

### Related Work / Table 1
- Recommend to add M2D-Resp (Niizumi et al., 2025), which demonstrates better performance than OPERA as a representation learning approach, to make the comparison more complete.

### Figure 1
- The figure is too small overall and the text is too small to read — improvements in size and font are required.
- The caption ("The framework takes textual respiratory prompts as input and generates synthetic lung sounds through a latent diffusion architecture.") does not match the figure content — correction is needed.

### Section 3 (Overall Picture of LungTTA)
- The term "fine-tuned" appears, but the overall picture is not clearly presented. State explicitly in the Section 3 introduction that LungTTA is a fine-tuned version of Stable Audio Open, to ensure a smooth reading flow.
- Explicitly specify which pre-trained models are used for which components (the DiT = Stable Audio Open is stated, but there is no mention of the VAE).

### Section 3.1 (Multiple Gaps in Explanation)

1. **Audio encoder**: Clarify the role of the audio encoder on the left side of the figure (is it the VAE encoder or something different?).
2. **Parameter count**: For "96.41M trainable parameters," specify the details (e.g., whether LoRA or only certain blocks are used).
3. **Loss computation order**: Clarify whether the loss is computed after watermarking.
4. **Flan-T5 output**: Clarify whether the tokenized prompt text produces a single vector or a sequence of vectors.
5. **Retrieval-Based Memory Bank — q**: Define what q is. The figure suggests it is the prompt text token, but since H = f_text(p), does q = p? If the prompt is a sequence, explain whether each token is replaced by the nearest neighbor exemplar — i.e., whether this amounts to mapping a freely written prompt to the closest prompt in the original dataset.
6. **Definition of k**: k is not explained — is it the prompt token sequence length?
7. **Formula for Z_mem**: The mathematical description is incomplete — is Z_mem = {e_{i1}, …, e_{ik}}? Complete the formulation.
8. **Z_meta**: No explanation is provided whatsoever — a description must be added.
9. **"the data" in Prompt-Based Conditioning**: Clarify whether "the data" refers to the training set. Also explain what "This setup lets the model follow the prompt while still being influenced by examples from the data" concretely means.

### PRISM — Definition and Consistency
- Clarify the relationship between the title label "A Domain-Aware Consistency Metric" and the defined acronym "Pulmonary Respiratory Integrity & Similarity Metric" — these are currently difficult to reconcile.
- Provide justification for: the choice of each w_i value, the use of MFCC for S_traj, the use of DTW for matching, and what Similarity is computed against. Each decision must be supported by prior literature or principled reasoning.

### Metric Validation (Sample Size)
- Validation on only 20 samples is statistically insufficient. Use the full test sets of each major existing dataset and report proper statistics.

### Section 4 (Experimental Setup)
- Present the overall flow: how fine-tuning was conducted and in what order evaluation was performed.
- Describe how audio samples of varying lengths are handled (within datasets and across batches).
- Explicitly state in the "Training" section that Stable Audio Open is used as the backbone.
- Describe how variable-length audio is handled in batching.
- Rename the section title to something more descriptive, such as "Experimental Setup."
- Add cross-references to the Appendix from the main text.

### Section 5
- Improve the opening description of the section. Clarify whether the items listed in "We evaluate LungTTA using a mix of objective metrics, listening studies, ablation experiments, and a downstream classification task" constitute metrics or evaluation procedures.

### Section 5.1
- Describe what experiment was conducted and state the value of the experiment clearly.

### Section 5.1.1
- Provide experimental details: how many samples/pairs were compared and whether the results are statistically meaningful. Report STD and/or CI. Also reconsider whether a sub-subsection number (5.1.1) is necessary.

### Section 5.1.2
- Clearly define "the baseline model" — is it Stable Audio Open fine-tuned straightforwardly? This is not stated anywhere.
- Provide a quantitative definition of "minimal change."

### Section 5.1.3
- State the rationale for this experimental setting.
- Consider using the OPERA benchmark, which enables more comprehensive evaluation.
- Add explanations and citations for "Classific augmentation" and "GAN-based" methods.

### Table 5
- Two entries under Specificity are bolded, despite LungTTA performing lower than GAN on this metric — only the GAN result should be bolded.

### Section 5.2
- Report the evaluation details: number of evaluators, whether they are clinicians, and the definition of N and N=32. Without these, the experimental value cannot be assessed.

---

> ### Author Response · Authors · 2026-07-04
> **Summary of revisions addressing Reviewer jADz’s comments on LungTTA’s novelty, methodology, notation, PRISM validation, experimental setup, evaluation reporting, and reproducibility.**
>
> Dear Reviewer,
>
> We thank the reviewer for the constructive comments. We are encouraged that the reviewer finds the problem setting promising and relevant to TMLR. The review has helped us improve the manuscript substantially, particularly in clarifying the contribution, methodology, notation, experimental design, and evaluation.
>
> In the revised manuscript, LungTTA is positioned as a respiratory-domain adaptation of the Stable Audio Open text-to-audio backbone. We explicitly separate inherited and introduced components. The pretrained audio autoencoder, T5-based text encoder, and diffusion transformer backbone are inherited from Stable Audio Open 1.0, while LungTTA adds respiratory-specific prompt construction, metadata conditioning, retrieval-guided memory conditioning, watermarking, PRISM, and a respiratory-audio evaluation pipeline.
>
> We have expanded the Method and Training sections. The revised manuscript now describes preprocessing, stereo waveform input at 44.1 kHz, fixed-length segments of 262,144 samples, latent encoding, T5-base prompt encoding, duration conditioning, retrieval memory with top-k=4, memory dropout, DiT configuration, AdamW optimization, and the STFT-based loss term. We also clarify that variable-length recordings are segmented or padded before batching, and that the audio encoder in Figure 1 is the pretrained audio autoencoder encoder.
>
> The conditioning notation has been revised. We now define `H`, `Z_mem`, and `Z_meta` before introducing `Z_cond = [H; Z_mem; Z_meta]`, and clarify that this denotes conditioning-level fusion rather than a fixed tensor operation. The retrieval query `q` is defined as a prompt-level embedding derived from the encoded prompt representation, not the raw prompt text or an individual token. Similarly, `k` is defined as the number of retrieved memory entries, not the prompt length. Retrieved entries are auxiliary conditioning tokens and do not replace prompt tokens, retrieve waveform audio, or copy training recordings.
>
> Following the reviewer’s suggestion, we revised the Related Work section and Table 1 to include M2D-Resp/M2D+Resp by Niizumi et al. (2025). We clarify that this work addresses respiratory representation learning, while LungTTA addresses controllable text-to-audio generation for synthetic respiratory audio. Figure 1 has also been redesigned with larger text, clearer labels, and a revised caption.
>
> The PRISM section has been revised. We now use one consistent name: PRISM: Pulmonary Respiratory Integrity and Similarity Metric. We clarify that PRISM compares generated respiratory recordings with matched reference recordings from the held-out evaluation set. We justify MFCC trajectory similarity, DTW alignment, respiratory rhythm, wheeze-like coherence, crackle-like transient structure, and respiratory band-energy distribution. The component weights are transparent heuristic choices rather than clinically optimized parameters. We add sensitivity analysis using equal weighting and leave-one-component-out variants, and expand PRISM validation with generated-audio versus held-out reference evaluation.
>
> The experimental section has been reorganized into a chronological flow covering dataset preparation, preprocessing, prompt construction, fine-tuning, generation, objective evaluation, human evaluation, ablation testing, watermark testing, and downstream augmentation. Section 4 is renamed Experimental Setup, with appendix cross-references for implementation details, datasets, hyperparameters, examples, and additional analyses.
>
> We also improve evaluation reporting. Objective metrics such as FAD, KL, IS, and PRISM are now separated from listening studies, expert assessment, ablation experiments, watermark testing, and downstream classification. We report sample counts, describe comparisons, include uncertainty estimates where appropriate, and define the baseline as the fine-tuned Stable Audio Open backbone without retrieval memory and watermarking.
>
> The downstream COVID-19 cough experiment is framed as a controlled augmentation benchmark, not as clinical evidence for cough-based diagnosis. We discuss shortcut-learning risks, avoid clinical overclaiming, and expand classical augmentation and GAN-based baselines, with details in the appendix.
>
> Finally, the subjective evaluation section reports the number of evaluators, samples, ratings per model, randomization, blinding, and whether expert raters were respiratory-health professionals. We clarify `N=32`, correct Table 5 so that only the best value in each column is highlighted, and revise Figure 3’s caption to avoid unfair attribution to architecture alone.
>
> Overall, these revisions present LungTTA more clearly as a respiratory-domain adaptation and evaluation framework built on Stable Audio Open, while addressing concerns regarding novelty, implementation clarity, notation, PRISM validation, experimental completeness, and reproducibility.

---

> > ### Comment · Reviewer_jADz · 2026-07-05
> > **Please upload the revised manuscript**
> >
> > Thank you for your comments.
> > We need the revised manuscript for continuing the review.
> > Thanks in advance.

---

> > > ### Author Response · Authors · 2026-07-05
> > > **Author Response**
> > >
> > > Thank you for your message. We are currently finalizing the revisions and will submit the revised manuscript as soon as possible. Thank you for your patience and consideration.

---

> > > > ### Comment · Reviewer_jADz · 2026-07-12
> > > > **Highlighted or tracked-changes version?**
> > > >
> > > > Thank you for the update. To facilitate the review, could you please upload a highlighted or tracked-changes version, as is customary for journal peer reviews? Thank you.

---

> > > > > ### Author Response · Authors · 2026-07-12
> > > > > **Author response**
> > > > >
> > > > > Thank you for your suggestion. We appreciate your feedback. We will upload a highlighted version of the revised manuscript later today. Thank you again for your helpful comment.

---

### Author Response · Authors · 2026-07-12
**Highlighted Version of the Revised Manuscript Submitted**

Dear Editors and Reviewers,

We sincerely thank you for your time, careful assessment, and valuable feedback. Your comments have helped us substantially improve the clarity, technical detail, experimental presentation, and overall positioning of the manuscript. We have now submitted a highlighted version of the revised manuscript.

The revisions include:

* Clarifying that LungTTA is a respiratory-domain adaptation of Stable Audio Open and clearly distinguishing the inherited backbone components from the contributions introduced in this work.
* Expanding the methodological and training details, including the fine-tuning procedure, model components, conditioning formulation, retrieval-based memory bank, metadata conditioning, parameter configuration, loss computation, and variable-length audio handling.
* Revising the mathematical notation and providing clearer definitions of the prompt, memory, and metadata conditioning representations.
* Improving the LungTTA pipeline figure with larger text, clearer labels, and a revised caption.
* Strengthening the motivation and formulation of PRISM, including clearer justification of its components and weighting choices, additional sensitivity analysis, controlled perturbation experiments, and expanded validation.
* Reorganising the experimental setup and improving the reporting of objective metrics, ablation studies, human and expert evaluations, watermark testing, and downstream classification experiments.
* Providing additional information regarding sample sizes, uncertainty estimates, evaluator backgrounds, randomisation, blinding, and evaluation procedures.
* Expanding the downstream evaluation to include an additional asthma-classification experiment and interpreting the findings more cautiously as controlled data-augmentation results rather than clinical evidence.
* Strengthening the discussion of watermarking, limitations, ethical considerations, reproducibility, and the intended research use of the generated respiratory audio.

We are also progressively adding the implementation code, evaluation scripts, and supporting documentation to the anonymous GitHub repository. The materials are being uploaded gradually while we carefully review the files and remove names, institutional references, local file paths, metadata, and any other information that could compromise the anonymity of the submission. We will continue expanding and organising the repository to support reproducibility while maintaining the requirements of double-blind review.

Due to the scope and page-length constraints of the current manuscript, several broader extensions are identified as future work. These include evaluation across more diverse clinical conditions and recording environments, additional downstream respiratory tasks, larger-scale PRISM validation across more diverse datasets, further sensitivity analysis of the retrieval-memory configuration, and continued improvements to prompt fidelity and clinically relevant respiratory structure.

We are very grateful for the constructive feedback and for the opportunity to revise and strengthen the work. We hope that the revised manuscript addresses the reviewers’ concerns and presents the contributions, evaluation, and limitations of LungTTA more clearly.

Sincerely,
The Authors